# Universal Stagewise Learning for Non-Convex Problems with Convergence on Averaged Solutions

**Zaiyi Chen** [*]
University of Science and Technology of China

**Zhuoning Yuan** [*]
University of Iowa

**Jinfeng Yi**
JD AI Research

**Bowen Zhou**
JD AI Research

**Enhong Chen**
University of Science and Technology of China

**Tianbao Yang**
University of Iowa

## Abstract

Although stochastic gradient descent (SGD) method and its variants (e.g., stochastic momentum methods, ADAGRAD) are algorithms of choice for solving non-convex problems (especially deep learning), big gaps still remain between the theory and the practice with many questions unresolved. For example, there is still a lack of theories of convergence for SGD and its variants that use stagewise step size and return an averaged solution in practice. In addition, theoretical insights of why adaptive step size of ADAGRAD could improve non-adaptive step size of SGD is still missing for non-convex optimization. This paper aims to address these questions and fill the gap between theory and practice. We propose a universal stagewise optimization framework for a broad family of **non-smooth non-convex** problems with the following key features: (i) at each stage any suitable stochastic convex optimization algorithms (e.g., SGD or ADAGRAD) that return an averaged solution can be employed for minimizing a regularized convex problem; (ii) the step size is decreased in a stagewise manner; (iii) an averaged solution is returned as the final solution. Our theoretical results of stagewise ADAGRAD exhibit its adaptive convergence, therefore shed insights on its faster convergence than stagewise SGD for problems with slowly growing cumulative stochastic gradients. To the best of our knowledge, this is the first work for addressing the unresolved issues of existing theories mentioned earlier. Besides theoretical contributions, our empirical studies show that our stagewise SGD and ADAGRAD improve the generalization performance of existing variants/implementations of SGD and ADAGRAD.

## 1 Introduction

Non-convex optimization has recently received increasing attention due to its popularity in emerging machine learning tasks, particularly for learning deep neural networks. One of the keys to the success of deep learning for big data problems is the employment of simple stochastic algorithms such as SGD or ADAGRAD (Krizhevsky et al., 2012; Dean et al., 2012). Analysis of these stochastic algorithms for non-convex optimization is an important and interesting research topic, which already attracts much attention from the community of theoreticians (Ghadimi & Lan, 2013; 2016; Yang et al., 2016; Davis & Drusvyatskiy, 2018b; Ward et al., 2018; Li & Orabona, 2018). However, one issue that has been largely ignored in existing theoretical analysis is that the employed algorithms in practice usually differ from their plain versions that are well understood in theory. Below, we will discuss several important *heuristics* used in practice for training deep neural networks and the *gap* between the practice and the theory to motivate this work.

First, a trick for setting the step size in training deep neural networks is to change it in a stagewise manner from a large value to a small value, i.e., a constant step size is used in a stage for a number of iterations and is decreased for the next stage (Krizhevsky et al., 2012; Ren et al., 2018). Although

---

[*]Equal contribution. Correspondence to: `tianbao-yang@uiowa.edu`

this trick has been adopted by most open-sourced libraries, e.g., Caffe (Jia et al., 2014), Tensor-Flow (Abadi et al., 2015), Pytorch (Paszke et al., 2017), it still lacks theoretical analysis to date for non-convex optimization. A related question that stands out is how and when to decrease the step size. On standard benchmark datasets for academic use such as CIFAR-10, CIFAR-100 (Krizhevsky et al.), people could follow the setting reported in previous studies to get a good result, which however might not work well for new datasets. Hence, a better solution to setting the stagewise step size with insights from theory would be much preferred. However, in the existing literature of theory for non-convex optimization (Ghadimi & Lan, 2013; Davis & Drusvyatskiy, 2018b), only strategies based on an iteratively decreasing step size or a small constant step size have been well analyzed. For example, the existing theory usually suggests an iteratively decreasing step size proportional to $1/\sqrt{t}$ at the $t$-th iteration or a small constant step size, e.g., proportional to $\epsilon^2$ with $\epsilon \ll 1$ for finding an $\epsilon$-stationary solution whose gradient's magnitude (in expectation) is small than $\epsilon$.

Second, the averaging heuristic is usually used in practice, i.e., an averaged solution is returned for prediction (Bottou, 2010), which could yield improved stability and generalization (Hardt et al., 2016). However, existing theory for many stochastic non-convex optimization algorithms only provides guarantee on a uniformly sampled solution or a non-uniformly sampled solution with *decreasing* probabilities for latest solutions (Ghadimi & Lan, 2013; Yang et al., 2016; Davis & Drusvyatskiy, 2018b). In particular, if an iteratively decreasing step size proportional to $1/\sqrt{t}$ at the $t$-th iteration is employed, the convergence guarantee was provided for a random solution that is non-uniformly selected from all iterates with a sampling probability proportional to $1/\sqrt{t}$ for the $t$-th iterate. This means that the latest solution always has the smallest probability to be selected as the final solution, which contradicts to the common wisdom. If a small constant step size is used, then usually a uniformly sampled solution is provided with convergence guarantee. However, both options are rarely used in practice and cannot justify the heuristic that returns the last solution.

A third common approach in practice is to use adaptive coordinate-wise step size of ADA-GRAD (Dean et al., 2012). Although adaptive step size has been well analyzed for convex problems (e.g., why and when it could yield faster convergence than SGD) (Duchi et al., 2011; Chen et al., 2018b), it still remains an mystery for non-convex optimization with missing insights from theory. Several recent studies have attempted to analyze ADAGRAD for non-convex problems (Ward et al., 2018; Li & Orabona, 2018; Chen et al., 2018a; Zou & Shen, 2018). Nonetheless, none of them are able to exhibit the adaptive convergence of ADAGRAD to data as in the convex case and its potential advantage over SGD for non-convex problems.

To overcome the shortcomings of existing theories for stochastic non-convex optimization, this paper analyzes new algorithms that employ some or all of these commonly used heuristics in a systematic framework, aiming to fill the gap between theory and practice. The main results and contributions are summarized below:

- We propose a universal stagewise optimization framework for solving a family of non-convex problems, i.e., weakly convex problems, which is broader than smooth non-convex problems and includes some non-smooth non-convex problems. At each stage, any suitable stochastic convex optimization algorithms (e.g., SGD, ADAGRAD) with a constant step size parameter can be employed for optimizing a regularized convex problem with a number of iterations, which usually return an averaged solution. The step size parameter is decreased in a stagewise manner following a polynomial decaying scheme.

- We analyze several variants of the proposed framework by employing different basic algorithms, including SGD, ADAGRAD, stochastic heavy-ball (SHB) method, and stochastic Nesterov's accelerated gradient (SNAG) method. We prove the convergence of their stagewise versions for an averaged solution that is randomly selected from all stagewise averaged solutions.

- To justify a heuristic approach that returns the last averaged solution in stagewise learning, we present and analyze a non-uniform sampling strategy over stagewise averaged solutions with sampling probabilities increasing as the stage number.

- Regarding the convergence results, for stagewise SGD, SHB, SNAG, we establish the same order of iteration complexity for finding a nearly stationary point as the existing theories of their non-stagewise variants. For stagewise ADAGRAD, we establish an adaptive convergence for finding a nearly stationary point, which is provably better than (stagewise) SGD, SHB, and SNAG when the cumulative growth of stochastic gradient is slow.

- Besides theoretical contributions, we also empirically verify the effectiveness of the proposed stagewise algorithms. In particular, our empirical studies show that (i) the stagewise ADAGRAD dramatically improves the generalization performance of existing variants of ADAGRAD, (ii) stagewise SGD, SHB, SNAG also outperform their plain variants with an iteratively decreasing step size; (iii) the proposed stagewise algorithms achieve similar if not better generalization performance than their heuristic variants implemented in existing libraries on standard benchmark datasets.

## 2    RELATED WORK

SGD for unconstrained smooth non-convex problems was first analyzed by Ghadimi & Lan (2013), who established an $O(1/\epsilon^4)$ iteration complexity for finding an $\epsilon$-stationary point $\mathbf{x}$ in expectation satisfying $\mathrm{E}[\|\nabla f(\mathbf{x})\|] \leq \epsilon$, where $f(\cdot)$ denotes the objective function. As mentioned earlier, the returned solution is either a uniformly sampled solution or a non-uniformly sampled one with sampling probabilities proportional to the decreasing step size. Similar results were established for the stochastic momentum variants of SGD (i.e., SHB, SNAG) by Yang et al. (2016); Ghadimi & Lan (2016). Recently, SGD was also analyzed for (constrained) weakly convex problems, whose objective function is non-convex and not necessarily smooth, by Davis & Drusvyatskiy (2018b). However, none of these studies provide results for algorithms that return an averaged solution, and these analyzed algorithms differ significantly from that used in practice for achiving the state-of-the-art results (Krizhevsky et al., 2012; Ren et al., 2018; Loshchilov & Hutter, 2017).

Although adaptive variants of SGD, e.g., ADAGRAD (Duchi et al., 2011), ADAM (Kingma & Ba, 2015; Reddi et al., 2018), were widely used for training deep neural networks, there are few studies on theoretical analysis of these algorithms for non-convex problems. Several recent studies attempted to analyze ADAGRAD for non-convex problems (Ward et al., 2018; Li & Orabona, 2018; Chen et al., 2018a; Zou & Shen, 2018). Although these studies have established an iteration complexity of $O(1/\epsilon^4)$ for different variants of ADAGRAD for finding an $\epsilon$-stationary solution of a stochastic non-convex optimization problem, none of them can exhibit the potential adaptive advantage of ADAGRAD over SGD as in the convex case. Besides that, these studies also suffer from the following shortcomings: (i) they all assume smoothness of the objective function, while we consider non-smooth and non-convex problems; (ii) their convergence is provided on a solution with minimum magnitude of gradient that is expensive to compute, though their results also imply a convergence on a random solution selected from all iterates with decreasing sampling probabilities. In contrast, these shortcomings do not exist in this paper. To the best of our knowledge, our result is the first one that explicitly shows that coordinate-wise adaptive step size could yield faster convergence than using non-adaptive step size for *non-smooth non-convex problems*, which is similar to that in the convex case and was observed in practice for deep learning (Dean et al., 2012).

The proposed stagewise algorithm is similar to several existing algorithms in design (Xu et al., 2017; Davis & Grimmer, 2017), which are originated from the proximal point algorithm (Rockafellar, 1976). I.e., at each stage a regularized convex subproblem is formed and then a stochastic algorithm is employed for optimizing the regularized subproblem inexactly with a number of iterations. Xu et al. (2017) used this idea for solving problems that satisfy a local error bound condition, aiming to achieve faster convergence than vanilla SGD. Davis & Grimmer (2017) are probably the first who analyzed this idea for solving non-smooth weakly convex problems. In these two papers SGD with decreasing step sizes for a strongly convex problem is employed at each stage for solving the regularized subproblem. Our stagewise algorithm is developed following the similar idea. The key differences from (Xu et al., 2017; Davis & Grimmer, 2017) are that (i) we focus on non-convex problems instead of convex problems considered in (Xu et al., 2017); (ii) we analyze a non-uniform sampling strategy with sampling probabilities increasing as the stage number to justify a heuristic approach that uses the last averaged solution for prediction, unlike the uniform sampling used in (Davis & Grimmer, 2017); (iii) we present a unified algorithmic framework and convergence analysis, which enable one to employ any suitable stochastic convex optimization algorithms at each stage. Importantly, it brings us several interesting variants including stagewise stochastic momentum methods and stagewise ADAGRAD. Finally, we note that a similar idea that adds a strongly convex term into the non-convex objective has been considered in several recent works (Allen-Zhu, 2017; Lan & Yang, 2018; Carmon et al., 2016). However, the key difference between these works and the present work is that they need to assume the objective function is smooth or has a smooth component. In addition, Allen-Zhu (2017); Lan & Yang (2018) consider finite-sum problems, and Carmon et al.

(2016) consider deterministic problems. In contrast, we consider more general stochastic problems without assuming the objective function is smooth.

## 3 PRELIMINARIES

The problem of interest in this paper is:

$$\min_{\mathbf{x} \in \Omega} \phi(\mathbf{x}) = \mathrm{E}_\xi[\phi(\mathbf{x}; \xi)], \tag{1}$$

where $\Omega \subseteq \mathbb{R}^d$ is a closed convex set, $\xi \in \mathcal{U}$ is a random variable, $\phi(\mathbf{x})$ and $\phi(\mathbf{x}; \xi)$ are non-convex functions, with the basic assumptions on the problem given in Assumption 1.

To state the convergence property of an algorithm for solving the above problem. We need to introduce some definitions. These definitions can be also found in related literature, e.g., Davis & Grimmer (2017); Davis & Drusvyatskiy (2018b). In the sequel, we let $\| \cdot \|$ denote an Euclidean norm, $[S] = \{1, \ldots, S\}$ denote a set, and $\delta_\Omega(\cdot)$ denote the indicator function of the set $\Omega$.

**Definition 1.** *(Fréchet subgradient) For a non-smooth and non-convex function $f(\cdot)$,*

$$\partial_F f(\mathbf{x}) = \{\mathbf{v} \in \mathbb{R}^d | f(\mathbf{y}) \geq f(x) + \mathbf{v}^\top (\mathbf{y} - \mathbf{x}) + o(\|\mathbf{y} - \mathbf{x}\|), \ \forall \mathbf{y} \in \mathbb{R}^d\}$$

*denotes the Fréchet subgradient of $f$.*

**Definition 2.** *(First-order stationarity) For problem (1), a point $\mathbf{x} \in \Omega$ is a first-order stationary point if $0 \in \partial_F(\phi + \delta_\Omega)(\mathbf{x})$, where $\delta_\Omega$ denotes the indicator function of $\Omega$. Moreover, a point $\mathbf{x}$ is said to be $\epsilon$-stationary if $dist(0, \partial_F(\phi + \delta_\Omega)(\mathbf{x})) \leq \epsilon$, where $dist$ denotes the Euclidean distance from a point to a set.*

**Definition 3.** *(Moreau Envelope and Proximal Mapping) For any function $f$ and $\lambda > 0$, the following function is called a Moreau envelope of $f$*

$$f_\lambda(\mathbf{x}) = \min_{\mathbf{z}} f(\mathbf{z}) + \frac{1}{2\lambda} \|\mathbf{z} - \mathbf{x}\|^2. \tag{2}$$

*Further, the optimal solution to the above problem denoted by*

$$prox_{\lambda f}(\mathbf{x}) = \arg\min_{\mathbf{z}} f(\mathbf{z}) + \frac{1}{2\lambda} \|\mathbf{z} - \mathbf{x}\|^2$$

*is called a proximal mapping of $f$.*

**Definition 4.** *(Weakly convex) A function $f$ is $\mu$-weakly convex ($\mu > 0$), if $f(\mathbf{x}) + \frac{\mu}{2} \|\mathbf{x}\|^2$ is convex.*

It is known that if $f(\mathbf{x})$ is $\mu$-weakly convex and $\gamma < \mu^{-1}$, then its Moreau envelope $f_\gamma(\mathbf{x})$ is $C^1$-smooth with the gradient given by (see e.g., Davis & Drusvyatskiy (2018b))

$$\nabla f_\gamma(\mathbf{x}) = \gamma^{-1}(\mathbf{x} - prox_{\gamma f}(\mathbf{x})).$$

The tool of Moreau envelope is introduced to measure the convergence for optimizing non-smooth and non-convex functions. A small norm of $\nabla f_\gamma(\mathbf{x})$ has an interpretation that $\mathbf{x}$ is close to a point that is nearly stationary. In particular for any $\mathbf{x} \in \mathbb{R}^d$, let $\widehat{\mathbf{x}} = prox_{\gamma f}(\mathbf{x})$, then we have (Davis & Drusvyatskiy, 2018b)

$$f(\widehat{\mathbf{x}}) \leq f(\mathbf{x}), \quad \|\mathbf{x} - \widehat{\mathbf{x}}\| = \gamma\|\nabla f_\gamma(\mathbf{x})\|, \quad dist(0, \partial f(\widehat{\mathbf{x}})) \leq \|\nabla f_\gamma(\mathbf{x})\|, \tag{3}$$

where the first inequality follows from the definition of $prox_{\lambda f}(\mathbf{x})$ such that $f(\widehat{\mathbf{x}}) + \frac{1}{2\lambda}\|\widehat{\mathbf{x}} - \mathbf{x}\|^2 \leq f(\mathbf{x})$ (Please see Appendix H for a detailed proof of (3)). This means that a point $\mathbf{x}$ satisfying $\|\nabla f_\gamma(\mathbf{x})\| \leq \epsilon$ is close to a point in distance of $O(\epsilon)$ that is $\epsilon$-stationary.

It is notable that for a non-smooth function $f(\cdot)$, there could exist a sequence of solutions $\{\mathbf{x}_k\}$ such that $\nabla f_\gamma(\mathbf{x}_k)$ converges while $dist(0, \partial f(\mathbf{x}_k))$ may not converge (Drusvyatskiy & Paquette, 2018). A simple example is to consider $\min_{x \in \mathbb{R}} |x|$. As long as $x \neq 0$, $dist(0, \partial f(x)) = 1 \neq 0$ no matter how close is $x$ to the stationary point 0. To handle such a challenging issue for non-smooth non-convex problems, we will follow existing works (Davis & Drusvyatskiy, 2018a; Drusvyatskiy & Paquette, 2018; Davis & Grimmer, 2017) to prove the near stationarity in terms of $\nabla f_\gamma(\mathbf{x})$. In the case when $f$ is smooth, $\|\nabla f_\gamma(\mathbf{x})\|$ is closely related to the magnitude of the gradient. In

particular, let us define the projected gradient below, which is used as a criterion for non-convex smooth optimization in the presence of constraints (Ghadimi & Lan, 2016; Allen-Zhu, 2017):

$$\mathcal{G}_\gamma(\mathbf{x}) = \frac{1}{\gamma}(\mathbf{x} - \text{prox}_{\gamma\delta_\Omega}(\mathbf{x} - \gamma\nabla f(\mathbf{x}))). \tag{4}$$

Note that the projected gradient becomes the normal gradient when the constraint $\mathbf{x} \in \Omega$ is absent. It was shown that when $f(\cdot)$ is smooth with $L$-Lipschitz continuous gradient (Drusvyatskiy & Lewis, 2016):

$$(1 - L\gamma)\|\mathcal{G}_\gamma(\mathbf{x})\| \le \|\nabla f_\gamma(\mathbf{x})\| \le (1 + L\gamma)\|\mathcal{G}_\gamma(\mathbf{x})\|, \forall \mathbf{x} \in \Omega. \tag{5}$$

Thus, the near stationarity in terms of $\nabla f_\gamma(\mathbf{x})$ implies the near stationarity in terms of $\mathcal{G}_\gamma(\mathbf{x})$ for a smooth function $f(\cdot)$ for a properly chosen $\gamma > 0$ (e.g., $\gamma = 1/(2L)$). In this work, we define $\phi_\gamma(\mathbf{x})$ as the Moreau envelope of $\phi(\mathbf{x}) + \delta_\Omega(\mathbf{x})$ as in (2) with $f(\mathbf{x})$ replaced by $\phi(\mathbf{x}) + \delta_\Omega(\mathbf{x})$ and study the convergence in terms of of $\nabla\phi_\gamma(\mathbf{x})$.

Now, we are ready to state the basic assumptions of the considered problem (1).

**Assumption 1.** *(i) There is a measurable mapping $g : \Omega \times \mathcal{U} \to \mathbb{R}$ such that $\text{E}_\xi[g(\mathbf{x}; \xi)] \in \partial_F\phi(\mathbf{x})$ for any $\mathbf{x} \in \Omega$; (ii) for any $\mathbf{x} \in \Omega$, $\text{E}[\|g(\mathbf{x}; \xi)\|^2] \le G^2$; (iii) there exists $\Delta_\phi > 0$ such that $\phi(\mathbf{x}) - \min_{\mathbf{z}\in\Omega} \phi(\mathbf{z}) \le \Delta_\phi$ for any $\mathbf{x} \in \Omega$; (iv) the objective function $\phi$ is $\mu$-weakly convex;*

**Remark:** Assumption 1(i), (ii) assume a stochastic subgradient is available for the objective function and its Euclidean norm square is bounded in expectation, which are standard assumptions for non-smooth optimization. Assumption 1(iii) assumes that the objective value with respect to the optimal value is bounded. Assumption 1(iv) assumes weak convexity of the objective function, which is weaker than assuming smoothness. Below, we present some examples of objective functions in machine learning that are weakly convex.

**Ex. 1: Smooth Non-Convex Functions.** If $\phi(\cdot)$ is a $L$-smooth function (i.e., its gradient is $L$-Lipschitz continuous), then it is $L$-weakly convex. This will include the objective function for neural networks with a smooth activation function (e.g., the sigmoid function) and a smooth loss function (e.g., softmax loss).

**Ex. 2: Convex and Smooth Composition.** Consider $\phi(\mathbf{x}; \xi) = h(c(\mathbf{x}; \xi))$ where $h(\cdot) : \mathbb{R}^m \to \mathbb{R}$ is closed convex and $M$-Lipschitz continuous, and $c(\mathbf{x}; \xi) : \mathbb{R}^d \to \mathbb{R}^m$ is nonlinear smooth mapping with $L$-Lipschitz continuous gradient. This class of functions has been considered in (Drusvyatskiy & Paquette, 2018) and it was proved that $\phi(\mathbf{x}; \xi)$ is $ML$-weakly convex. An interesting example is phase retrieval, where $\phi(\mathbf{x}; a, b) = |(\mathbf{x}^\top \mathbf{a})^2 - b|$. Another example related to deep learning is that if $c(\mathbf{x}; \xi)$ denotes the mapping function of a neural network parameterized by $\mathbf{x}$ with a smooth activation function and $h$ denotes a non-smooth Lipschitz continuous convex loss function (e.g., hinge loss), then the resulting loss $h(c(\mathbf{x}; \xi))$ is a weakly convex function of $\mathbf{x}$. More examples of this class can be found in (Davis & Drusvyatskiy, 2018a).

## 4 STAGEWISE OPTIMIZATION: ALGORITHMS AND CONVERGENCE

In this section, we will present the proposed stagewise algorithms and their convergence results. We will first present a Meta algorithmic framework highlighting the key features of the proposed algorithms and then present several variants of the Meta algorithm.

The Meta algorithmic framework is described in Algorithm 1. There are several key features that differentiate Algorithm 1 from existing stochastic algorithms that come with theoretical guarantee. First, the algorithm is run with multiple stages. At each stage, a basic stochastic algorithm (SA) is called to optimize a regularized problem $f_s(\mathbf{x})$ inexactly that consists of the original objective function and a quadratic term, which is guaranteed to be convex due to the weak convexity of $\phi$ and $\gamma < \mu^{-1}$. The convexity of $f_s$ allows one to employ any suitable existing stochastic algorithms (cf. Theorem 1) that have convergence guarantee for convex problems. The returned solution from the $(s - 1)$-th stage is used as a reference point for constructing $f_s$ and as an initial solution for warm-start. It is notable that SA usually returns an averaged solution $\mathbf{x}_s$ at each stage. Second, a decreasing sequence of step size parameters $\eta_s$ is used. At each stage, the SA uses a constant step size parameter $\eta_s$ and runs the updates for a number of $T_s$ iterations. We do not initialize $T_s$ as it might be adaptive to the data as in stagewise ADAGRAD. The setup of $\eta_s$ and $T_s$ will depend on the specific choice of SA, which will be exhibited later for different variants.

---
**Algorithm 1** A Meta Stagewise Algorithm: Stagewise-SA

---
1: **Initialize:** a sequence of decreasing step size parameters $\{\eta_s\}$, $\mathbf{x}_0 \in \Omega$, $\gamma < \mu^{-1}$
2: **for** $s = 1, \ldots, S$ **do**
3:     Let $f_s(\cdot) = \phi(\cdot) + \frac{1}{2\gamma}\|\cdot - \mathbf{x}_{s-1}\|^2$
4:     $\mathbf{x}_s = \text{SA}(f_s, \mathbf{x}_{s-1}, \eta_s, T_s)$            $\diamond \mathbf{x}_s$ is usually an averaged solution
5: **end for**

---

To illustrate that Algorithm 1 is a universal framework such that any suitable SA algorithm can be employed, we present the following result by assuming that SA has an appropriate convergence for a convex problem.

**Theorem 1.** *Let $f(\cdot)$ be a convex function, $\mathbf{x}_* = \arg\min_{\mathbf{x}\in\Omega} f(\mathbf{x})$ and $\Theta$ denote some problem dependent constants. Suppose for $\mathbf{x}_+ = \text{SA}(f, \mathbf{x}_0, \eta, T)$, we have*

$$\text{E}[f(\mathbf{x}_+) - f(\mathbf{x}_*)] \leq \varepsilon_1(\eta, T, \Theta)\|\mathbf{x}_0 - \mathbf{x}_*\|_2^2 + \varepsilon_2(\eta, T, \Theta)(f(\mathbf{x}_0) - f(\mathbf{x}_*)) + \varepsilon_3(\eta, T, \Theta). \quad (6)$$

*Under Assumption 1(i), (iii) and (iv), by running Algorithm 1 with $\gamma = 1/(2\mu)$, and with $\eta_s, T_s$ satisfying $\varepsilon_1(\eta_s, T_s, \Theta) \leq 1/(48\gamma), \varepsilon_2(\eta_s, T_s, \Theta) \leq 1/2, \varepsilon_3(\eta_s, T_s, \Theta) \leq c_3/s$ for some $c_3 > 0$, we have*

$$\text{E}\big[\|\nabla\phi_\gamma(\mathbf{x}_\tau)\|^2\big] \leq \frac{32\Delta_\phi(\alpha+1)}{\gamma(S+1)} + \frac{48c_3(\alpha+1)}{\gamma(S+1)},$$

*where $\tau$ is randomly selected from $\{0, \ldots, S\}$ with probabilities $p_\tau \propto (\tau+1)^\alpha, \alpha \geq 1$.*

**Remark:** It is notable that the convergence guarantee is provided on a stagewise average solution $\mathbf{x}_\tau$. To justify a heuristic approach that returns the final average solution for prediction, we analyze a new sampling strategy that samples a solution among all stagewise average solutions with sampling probabilities increasing as the stage number increases. This sampling strategy is better than uniform sampling strategy or a strategy with decreasing sampling probabilities in the existing literature. The convergence upper bound in (6) of SA covers the results of a broad family of stochastic convex optimization algorithms. When $\varepsilon_2(\eta_s, T_s, \Theta) = 0$ (as in SGD), the upper bound can be improved by a constant factor. Moreover, we do not optimize the value of $\gamma$. Indeed, any $\gamma < 1/\mu$ will work, which only has an effect on constant factor in the convergence upper bound.

Next, we present several variants of the Meta algorithm by employing SGD, ADAGRAD, and stochastic momentum methods as the basic SA algorithm, to which we refer as stagewise SGD, stagewise ADAGRAD, and stagewise stochastic momentum methods, respectively. It is worth mentioning that one can follow similar analysis to analyze other stagewise algorithms by using their basic convergence for stochastic convex optimization, including RMSProp (Mukkamala & Hein, 2017), AMSGrad (Reddi et al., 2018), stochastic alternating direction methods of multipliers (ADMM) (Ouyang et al., 2013; Suzuki, 2013).

## 4.1 STAGEWISE SGD

In this subsection, we analyze the convergence of stagewise SGD, in which SGD shown in Algorithm 3 in the Appendix is employed in the Meta framework. Besides Assumption 1, we impose the following bounded domain assumption in this subsection.

**Assumption 2.** *There exists $D > 0$ such that $\|\mathbf{x} - \mathbf{y}\| \leq D$ for any $\mathbf{x}, \mathbf{y} \in \Omega$.*

It is worth mentioning that bounded domain assumption is imposed for simplicity, which is usually assumed in convex optimization. For machine learning problems, one usually imposes some bounded norm constraint to achieve a regularization. Nevertheless, the bounded domain assumption is not essential for the proposed algorithm. We present a more involved analysis in subsection 4.3 for unbounded domain $\Omega = \mathbb{R}^d$. The following is a standard basic convergence result of SGD (Zinkevich, 2003), which clearly satisfies the bound in (6).

**Lemma 1.** *For Algorithm 3, assume that $f(\cdot)$ is convex and $\text{E}\|\mathbf{g}_t\|^2 \leq G^2, \forall t$, then for any $\mathbf{x} \in \Omega$*

$$\text{E}[f(\widehat{\mathbf{x}}_T) - f(\mathbf{x})] \leq \frac{\|\mathbf{x} - \mathbf{x}_0\|^2}{2\eta(T+1)} + \frac{\eta G^2}{2}$$

The following theorem exhibits the convergence of stagewise SGD.

---

**Algorithm 2** ADAGRAD($f, \mathbf{x}_0, \eta, *$)

1: **Initialize:** $\mathbf{x}_1 = \mathbf{x}_0$, $\mathbf{g}_{1:0} = []$, $H_0 \in \mathbb{R}^{d \times d}$
2: **while** $T$ does not satisfy the condition in Theorem 3 **do**
3:    Compute a stochastic subgradient $\mathbf{g}_t$ for $f(\mathbf{x}_t)$
4:    Update $g_{1:t} = [g_{1:t-1}, \mathbf{g}(\mathbf{x}_t)]$, $s_{t,i} = \|g_{1:t,i}\|_2$
5:    Set $H_t = H_0 + \mathrm{diag}(\mathbf{s}_t)$ and $\psi_t(\mathbf{x}) = \frac{1}{2}(\mathbf{x} - \mathbf{x}_1)^\top H_t(\mathbf{x} - \mathbf{x}_1)$
6:    Let $\mathbf{x}_{t+1} = \arg\min_{\mathbf{x} \in \Omega} \eta \mathbf{x}^\top \left(\frac{1}{t}\sum_{\tau=1}^t \mathbf{g}_\tau\right) + \frac{1}{t}\psi_t(\mathbf{x})$
7: **end while**
8: **Output:** $\widehat{\mathbf{x}}_T = \sum_{t=1}^T \mathbf{x}_t / T$

---

**Theorem 2.** *Suppose Assumption 1 and 2 hold. By setting $\gamma = 1/(2\mu)$, $\eta_s = \eta_0/s$, $T_s = 12\gamma s/\eta_0$ where $\eta_0 > 0$ is a free parameter, then stagewise* SGD *(Algorithm 1 employing* SGD*) guarantees that*

$$\mathrm{E}\big[\|\nabla\phi_\gamma(\mathbf{x}_\tau)\|^2\big] \leq \frac{16\mu\Delta_\phi(\alpha+1)}{S+1} + \frac{24\mu\eta_0\hat{G}^2(\alpha+1)}{(S+1)},$$

*where $\hat{G}^2 = 2G^2 + 2\gamma^{-2}D^2$, and $\tau$ is similarly defined as in Theorem 1.*

**Remark:** To find a solution with $\mathrm{E}\big[\|\nabla\phi_\gamma(\mathbf{x}_\tau)\|^2\big] \leq \epsilon^2$, we can set $S = O(1/\epsilon^2)$ and the total iteration complexity $\sum_{s=1}^S T_s$ is in the order of $O(1/\epsilon^4)$. The above theorem is essentially a corollary of Theorem 1 by applying Lemma 1 to $f_s(\cdot)$ at each stage. We present a complete proof in the appendix.

### 4.2 STAGEWISE ADAGRAD

One of the main contributions of the present work is to develop a variant of ADAGRAD with adaptive convergence to data for stochastic non-convex optimization. In this subsection, we analyze stagewise ADAGRAD and establish its adaptive complexity. In particular, we consider the Meta algorithm that employs ADAGRAD in Algorithm 2 to optimize each $f_s$. The key difference of stagewise ADAGRAD from stagewise SGD is that the number of iterations $T_s$ at each stage is adaptive to the history of learning. It is this adaptiveness that makes the proposed stagewise ADAGRAD achieve adaptive convergence. It is worth noting that such adaptive scheme has been also considered in (Chen et al., 2018b) for solving stochastic strongly convex problems. In contrast, we consider stochastic weakly convex problems. Similar to previous analysis of ADAGRAD (Duchi et al., 2011; Chen et al., 2018b), we assume $\|g(\mathbf{x};\xi)\|_\infty \leq G, \forall \mathbf{x} \in \Omega$ in this subsection. Note that this is stronger than Assumption 1 (ii). We formally state this assumption required in this subsection below.

**Assumption 3.** $\|g(\mathbf{x};\xi)\|_\infty \leq G$ *for any* $\mathbf{x} \in \Omega$.

The convergence analysis of stagewise ADAGRAD is build on the following lemma, which is attributed to Chen et al. (2018b) with a proof sketch provided in the Appendix.

**Lemma 2.** *Let $f(\mathbf{x})$ be a convex function, $H_0 = GI$ with $G \geq \max_t \|\mathbf{g}_t\|_\infty$, and iteration number $T$ satisfy $T \geq M \max\{\frac{G+\max_i \|g_{1:T,i}\|}{2c}, c\sum_{i=1}^d \|g_{1:T,i}\|\}$ for some $c > 0$. Algorithm 2 returns an averaged solution $\widehat{\mathbf{x}}_T$ such that*

$$\mathrm{E}[f(\widehat{\mathbf{x}}_T) - f(\mathbf{x}_*)] \leq \frac{c}{M\eta}\|\mathbf{x}_0 - \mathbf{x}_*\|^2 + \frac{\eta}{Mc}, \tag{7}$$

*where $\mathbf{x}_* = \arg\min_{\mathbf{x} \in \Omega} f(\mathbf{x})$, $g_{1:t} = (\mathbf{g}(\mathbf{x}_1), \ldots, \mathbf{g}(\mathbf{x}_t))$ and $g_{1:t,i}$ denotes the $i$-th row of $g_{1:t}$.*

The convergence property of stagewise ADAGRAD is described by following theorem.

**Theorem 3.** *Suppose Assumption 1(i), (iii), (iv), Assumption 2 and Assumption 3 hold. By setting $\gamma = 1/(2\mu)$, $\eta_s = \eta_0/\sqrt{s}$, $T_s \geq M_s \max\{(\hat{G} + \max_i \|g_{1:T_s,i}^s\|)/(2c), c\sum_{i=1}^d \|g_{1:T_s,i}^s\|\}$ where $\eta_0, c > 0$ are free parameters, and $M_s\eta_s \geq 24\gamma c$, then stagewise* ADAGRAD *guarantees that*

$$\mathrm{E}[\|\nabla\phi_\gamma(\mathbf{x}_\tau)\|^2] \leq \frac{16\mu\Delta_\phi(\alpha+1)}{S+1} + \frac{4\mu^2\eta_0^2(\alpha+1)}{c^2(S+1)},$$

*where $\hat{G} = G + \gamma^{-1}D$, $g_{1:t,i}^s$ denotes the cumulative stochastic gradient of the $i$-th coordinate at the $s$-th stage, and $\tau$ is similarly defined as in Theorem 1.*

**Remark:** Note that the free parameter $c$ is introduced to balance the two terms in the lower bound of $T_s$ due to that $(\hat{G} + \max_i \|g_{1:T_s,i}^s\|) \propto \hat{G}\sqrt{T_s}$ and $\sum_{i=1}^d \|g_{1:T_s,i}^s\| \propto d\hat{G}\sqrt{T_s}$ have different orders when $d$ is very large. One way to balance these two terms is to set $c$ in the order of $\sqrt{1/d}$, resulting $O(d/(S+1))$ for the second term in the above convergence bound. Another way is to choose $c$ in the $s$-th stage such that the two terms in the max of the lower bound of $T_s$ match each other. One can derive a similar order of $O(d/(S+1))$ for the second term in the above convergence bound. It is obvious that the total number of iterations $\sum_{s=1}^S T_s$ is adaptive to the data. Next, let us present more discussion on the iteration complexity. Note that $M_s = O(\sqrt{s})$ by setting $c$ as a constant. By the boundness of stochastic gradient $\|g_{1:T_s,i}^s\| \leq O(\sqrt{T_s})$, therefore $T_s$ in the order of $O(s)$ will satisfy the condition in Theorem 3. Thus, in the worst case the iteration complexity for finding $\mathrm{E}[\|\nabla\phi_\gamma(\mathbf{x}_\tau)\|^2] \leq \epsilon^2$ is in the order of $\sum_{s=1}^S O(s) \leq O(1/\epsilon^4)$. To show the potential advantage of adaptive step size as in the convex case, let us consider a good case when the cumulative growth of stochastic gradient is slow, e.g., assuming $\|g_{1:T_s,i}^s\| \leq O(T_s^\alpha)$ with $\alpha < 1/2$. Then $T_s = O(s^{1/(2(1-\alpha))})$ will work, and then the total number of iterations $\sum_{s=1}^S T_s \leq S^{1+1/(2(1-\alpha))} \leq O(1/\epsilon^{2+1/(1-\alpha)})$, which is better than $O(1/\epsilon^4)$.

### 4.3 STAGEWISE STOCHASTIC MOMENTUM (SM) METHODS

Finally, we present stagewise stochastic momentum (SM) methods and their convergence results. In the literature, there are two popular variants of stochastic momentum methods, namely, stochastic heavy-ball method (SHB) and stochastic Nesterov's accelerated gradient method (SNAG). Both methods have been used for training deep neural networks (Krizhevsky et al., 2012; Sutskever et al., 2013), and have been analyzed by (Yang et al., 2016) for non-convex optimization. To contrast with the results in (Yang et al., 2016), we will consider the same unified stochastic momentum methods that subsume SHB, SNAG and SGD as special cases when $\Omega = \mathbb{R}^d$. The updates are presented in Algorithm 4 in the Appendix. There are two additional parameters: $\beta \in (0,1)$ is the momentum parameter and $\rho$ is a parameter that can vary between $[0, 1/(1-\beta)]$. By changing the value of $\rho$, we can obtain the three variants, SHB ($\rho = 0$), SNAG ($\rho = 1$) and SGD ($\rho = 1/(1-\beta)$). Due to the limit of space, we only present the convergence of stagewise SM methods below.

**Theorem 4.** *Suppose Assumption 1 holds. By setting $\gamma = 1/(2\mu)$, $\eta_s = (1-\beta)\gamma/(96s(\rho\beta+1))$, $T_s \geq 2304(\rho\beta+1)s$, with $\tau$ similarly defined as in Theorem 1 stagewise SM methods guarantee*

$$\mathrm{E}[\|\nabla\phi_\gamma(\mathbf{x}_\tau)\|^2] \leq \frac{16\mu\Delta_\phi(\alpha+1)}{S+1} + \frac{G^2(\beta+48(2\rho\beta+1)(1-\beta))(\alpha+1)}{96(\rho\beta+1)(1-\beta)(S+1)}.$$

**Remark:** The bound in the above theorem is in the same order as that in Theorem 2. The total iteration complexity for finding a solution $\mathbf{x}_\tau$ with $\mathrm{E}[\|\nabla\phi_\gamma(\mathbf{x}_\tau)\|^2] \leq \epsilon^2$ is $O(1/\epsilon^4)$ similar to that achieved in (Yang et al., 2016).

## 5 EXPERIMENTS

In this section, we present some empirical results to verify the effectiveness of the proposed stagewise algorithms. We use two benchmark datasets, namely CIFAR-10 and CIFAR-100 (Krizhevsky et al.) for our experiments. We implement the proposed stagewise algorithms in TensorFlow. We compare different algorithms for learning ResNet-20 (He et al., 2016) with batch normalization (Ioffe & Szegedy, 2015) adopted after each convolution and before ReLu activation.

**Baselines.** We compare the proposed stagewise algorithms with their variants implemented in TensorFlow. It is notable that ADAGRAD has a step size (aka learning rate) parameter [1], which is a constant in theory (Li & Orabona, 2018; Chen et al., 2018a; Zou & Shen, 2018). However, in the deep learning community a heuristic fixed frequency decay scheme for the step size parameter is commonly adopted (Ren et al., 2018; Wilson et al., 2017). We thus compare two implementations of ADAGRAD - one with a constant learning rate parameter and another one with a fixed frequency decay scheme, which are referred to as ADAGRAD (theory) and ADAGRAD (heuristic). For each baseline algorithms of SGD, SHB, SNAG, we also implement two versions - a theory version with iteratively decreasing size $\eta_0/\sqrt{t}$ suggested by previous theories and a heuristic approach with fixed frequency decay scheme used in practice, using (theory) and (heuristic) to indicate them. The fixed

---

[1] note it is not equivalent to the step size in SGD.

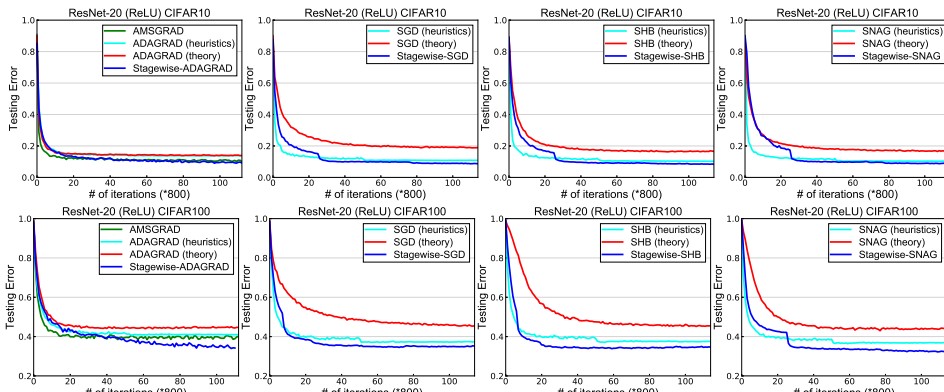

Figure 1: Comparison of Testing Error on CIFAR-10 (top) and CIFAR-100 (bottom).

frequency decay scheme used in the heuristic variants is similar as that in (He et al., 2016), i.e., the step size parameter is decreased by 10 at 40k, 60k iterations. We also compare stagwise ADAGRAD with AMSGrad (Reddi et al., 2018) - a corrected version of Adam.

**Parameters.** The stagwise step size $\eta_s = \eta_0/\sqrt{s}$ is used in stagwise ADAGRAD, the number of iterations $T_s$ in stagwise ADAGRAD is set according to Theorem 3 with some simplifications for dealing with unknown $\hat{G}$, in particular we set $T_s$ to the smallest value larger than $T_0\sqrt{s \max_i \|g_{1:T_s,i}^s\| \sum_i \|g_{1:T_s,i}^s\|}$. For stagwise SGD, SHB, SNAG, the stagwise step size and iteration number is set to $\eta_s = \eta_0/s$ and $T_s = T_0s$, respectively. The involved parameters in the compared algorithms are tuned for the best performance, including the initial step size parameter $\eta_0$ of all algorithms, the value of $T_0$ and $\gamma$ for our stagwise algorithms.

**Results.** We consider two settings - with/without an $\ell_2$ norm regularization on weights. For comparison, we evaluate the training error and testing error of obtained solutions in the process of training. For our stagwise algorithms, the evaluation is done based on the current averaged solution, and for other baselines the evaluation is done based on the current solution. Due to the limit of space, we only show the results of testing error on CIFAR-10 and CIFAR-100 for the setting without regularization in Figure 1. All results are included in the Appendix. From all results, we have several observations. (i) The proposed stagwise algorithms perform much better in terms of testing error than the existing theoretical versions reported in the literature (marked with theory in the legend). This indicates the proposed stagwise step size scheme is better than iteratively decreasing step size scheme. (ii) The proposed stagwise algorithms achieve similar and sometimes even better testing error than the heuristic approaches with a fixed frequency decay scheme used in practice. However, the heuristic approaches usually give smaller training error. This seems to indicate that the proposed algorithms are less vulnerable to the overfitting. In another word, the proposed algorithms have smaller generalization error, i.e., the difference between the testing error and the training error.

## 6 CONCLUSION & FUTURE WORK

In this paper, we have proposed a universal stagwise learning framework for solving stochastic non-convex optimization problems, which employs well-known tricks in practice that have not been well analyzed theoretically. We provided theoretical convergence results for the proposed algorithms for non-smooth non-convex optimization problems. We also established an adaptive convergence of a stochastic algorithm using data adaptive coordinate-wise step size of ADAGRAD, and exhibited its faster convergence than non-adaptive stepsize when the growth of cumulative stochastic gradients is slow similar to that in the convex case. For future work, one may consider developing more variants of the proposed meta algorithm, e.g., stagwise AMSGrad, stagwise RMSProp, etc. We will also consider the empirical studies on the large-scale ImageNet data set.

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

---

**Algorithm 3** $\text{SGD}(f, \mathbf{x}_0, \eta, T)$

---
    **for** $t = 0, \dots, T$ **do**
        Compute a stochastic subgradient $\mathbf{g}_t$ for $f(\mathbf{x}_t)$.
        $\mathbf{x}_{t+1} = \Pi_\Omega[\mathbf{x}_t - \eta\mathbf{g}_t]$
    **end for**
    **Output**: $\widehat{\mathbf{x}}_T = \sum_{t=0}^{T} \mathbf{x}_t/(T+1)$

---

**Algorithm 4** Unified Stochastic Momentum Methods: $\text{SUM}(f, \mathbf{x}_0, \eta, T)$

---
    **Set** parameters: $\rho \geq 0$ and $\beta \in (0,1)$.
    **for** $t = 0, \dots, T$ **do**
        Compute a stochastic subgradient $\mathbf{g}_t$ for $f(\mathbf{x}_t)$.
        $\mathbf{y}_{t+1} = \mathbf{x}_t - \eta\mathbf{g}_t$
        $\widehat{\mathbf{y}}_{t+1} = \mathbf{x}_t - \rho\eta\mathbf{g}_t$
        $\mathbf{x}_{t+1} = \mathbf{y}_{t+1} + \beta(\widehat{\mathbf{y}}_{t+1} - \widehat{\mathbf{y}}_t)$
    **end for**
    **Output**: $\widehat{\mathbf{x}}_T = \sum_{t=0}^{T} \mathbf{x}_t/(T+1)$

---

## A    MORE EXPERIMENTAL RESULTS

In this section, we present more experimental results. Comparison of training and testing error in the two settings (w/o regularization) on the two data sets are plotted in Figure 2, 3, 4, 5. We also report the final testing error (after running 80k iterations) of different algorithms in the two settings on the two datasets in Table 1. For parameter tuning, the initial step sizes of all algorithms are tuned in $\{0.1, 0.3, 0.5, 0.7, 0.9\}$. The value of $\gamma$ of stagewise algorithms is tuned in $\{1, 10, 100, 500, 1000, 1500, 2000, 3000\}$. The initial value $T_0$ for stagewise SGD, SHB, SNAG is tuned in $\{10, 100, 1\text{k}, 5\text{k}, 6\text{k}, 7\text{k}, 10\text{k}, 20\text{k}\}$, and that for stagewise ADAGRAD is tuned in $\{1, 10, 15, 20, 25, 50, 100\}$.

## B    PROOF OF THEOREM 1

*Proof.* Below, we use $\text{E}_s$ to denote expectation over randomness in the $s$-th stage given all history before $s$-th stage. Define

$$\mathbf{z}_s = \arg\min_{\mathbf{x}\in\Omega} f_s(\mathbf{x}) = \text{prox}_{\gamma(\phi+\delta_\Omega)}(\mathbf{x}_{s-1}) \tag{8}$$

Then $\nabla\phi_\gamma(\mathbf{x}_{s-1}) = \gamma^{-1}(\mathbf{x}_{s-1} - \mathbf{z}_s)$. By applying the convergence bound of SA to $f_s(\mathbf{x})$, we have

$$\text{E}_s[f_s(\mathbf{x}_s) - f_s(\mathbf{z}_s)] \leq \underbrace{\varepsilon_1(\eta_s, T_s, \Theta)\|\mathbf{x}_{s-1} - \mathbf{z}_s\|_2^2 + \varepsilon_2(\eta_s, T_s, \Theta)(f_s(\mathbf{x}_{s-1}) - f_s(\mathbf{z}_s)) + \varepsilon_3(\eta_s, T_s, \Theta)}_{\mathcal{E}_s}.$$

It then follows that

$$\text{E}_s\left[\phi(\mathbf{x}_s) + \frac{1}{2\gamma}\|\mathbf{x}_s - \mathbf{x}_{s-1}\|^2\right] \leq f_s(\mathbf{z}_s) + \mathcal{E}_s \leq f_s(\mathbf{x}_{s-1}) + \mathcal{E}_s$$

$$\leq \phi(\mathbf{x}_{s-1}) + \mathcal{E}_s \tag{9}$$

On the other hand, we have that

$$\begin{aligned}
\|\mathbf{x}_s - \mathbf{x}_{s-1}\|^2 &= \|\mathbf{x}_s - \mathbf{z}_s + \mathbf{z}_s - \mathbf{x}_{s-1}\|^2 \\
&= \|\mathbf{x}_s - \mathbf{z}_s\|^2 + \|\mathbf{z}_s - \mathbf{x}_{s-1}\|^2 + 2\langle\mathbf{x}_s - \mathbf{z}_s, \mathbf{z}_s - \mathbf{x}_{s-1}\rangle \\
&\geq (1 - \alpha_s^{-1})\|\mathbf{x}_s - \mathbf{z}_s\|^2 + (1 - \alpha_s)\|\mathbf{x}_{s-1} - \mathbf{z}_s\|^2
\end{aligned}$$

Table 1: Comparison of Final Testing Error (%) on CIFAR-10 and CIFAR-100 Datasets

| | CIFAR-10 | | CIFAR-100 | |
|---|---|---|---|---|
| Algorithms | with reg. | without reg. | with reg. | without reg. |
| SGD (theory) | 16.25 | 19.18 | 43.51 | 45.78 |
| SGD (heuristic) | 8.34 | 10.81 | 33.67 | 37.19 |
| Stagewise-SGD | **8.34** | **9.01** | **32.25** | **34.95** |
| SHB (theory) | 15.67 | 16.55 | 39.15 | 46.23 |
| SHB (heuristic) | 8.58 | 10.28 | 33.30 | 37.56 |
| Stagewise-SHB | **8.30** | **8.61** | **32.85** | **34.49** |
| SNAG (theory) | 17.64 | 16.76 | 39.34 | 44.21 |
| SNAG (heuristic) | 8.85 | 10.34 | 33.89 | 36.84 |
| Stagewise-SNAG | **8.00** | **8.93** | **31.42** | **33.29** |
| AMSGrad | 10.76 | 11.13 | 38.62 | 39.96 |
| AdaGrad (theory) | 12.11 | 13.96 | 39.09 | 44.49 |
| AdaGrad (heuristic) | 10.71 | 13.80 | 37.04 | 41.06 |
| Stagewise-AdaGrad | **9.09** | **9.51** | **33.95** | **34.62** |

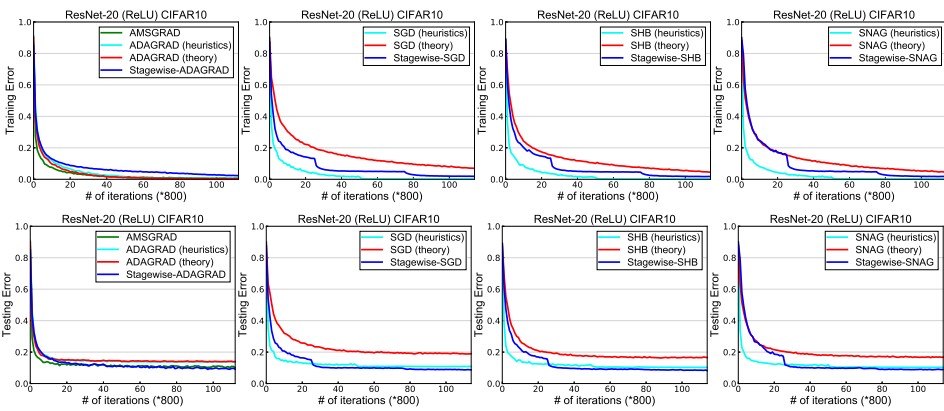

Figure 2: Comparison of Training Error (Top) and Testing Error (bottom) on CIFAR-10 without Regularization.

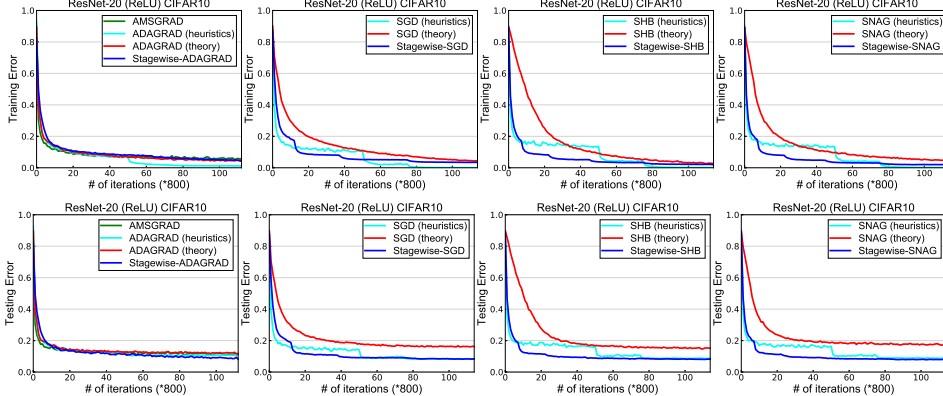

Figure 3: Comparison of Training Error (Top) and Testing Error (bottom) on CIFAR-10 with Regularization. The regularization parameter is set $5e - 4$.

where the inequality follows from the Young's inequality with $0 < \alpha_s < 1$. Combining the above inequality with (9) we have that

$$\mathrm{E}_s\left[\frac{(1-\alpha_s)}{2\gamma}\|\mathbf{x}_{s-1}-\mathbf{z}_s\|^2\right] \leq \mathrm{E}_s\left[\phi(\mathbf{x}_{s-1})-\phi(\mathbf{x}_s)+\frac{(\alpha_s^{-1}-1)}{2\gamma}\|\mathbf{x}_s-\mathbf{z}_s\|^2+\mathcal{E}_s\right]$$

$$\leq \mathrm{E}_s\left[\phi(\mathbf{x}_{s-1})-\phi(\mathbf{x}_s)+\frac{(\alpha_s^{-1}-1)}{\gamma(\gamma^{-1}-\mu)}(f_s(\mathbf{x}_s)-f_s(\mathbf{z}_s))+\mathcal{E}_s\right]$$

$$\leq \mathrm{E}_s\left[\phi(\mathbf{x}_{s-1})-\phi(\mathbf{x}_s)+\frac{\alpha_s^{-1}-\gamma\mu}{(1-\gamma\mu)}\mathcal{E}_s\right] \leq \mathrm{E}_s\left[\phi(\mathbf{x}_{s-1})-\phi(\mathbf{x}_s)\right]$$

$$+\mathrm{E}_s\left[\frac{\alpha_s^{-1}-\gamma\mu}{(1-\gamma\mu)}\{\varepsilon_1(\eta_s,T_s,\Theta)\|\mathbf{x}_{s-1}-\mathbf{z}_s\|^2+\varepsilon_2(\eta_s,T_s,\Theta)(f_s(\mathbf{x}_{s-1})-f_s(\mathbf{z}_s))+\varepsilon_3(\eta_s,T_s,\Theta)\}\right],$$

14

(10)

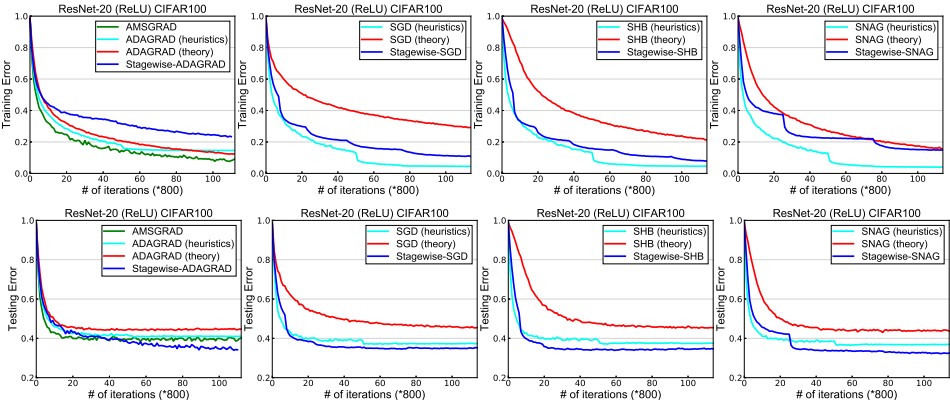

Figure 4: Comparison of Training Error (Top) and Testing Error (bottom) on CIFAR-100 without Regularization.

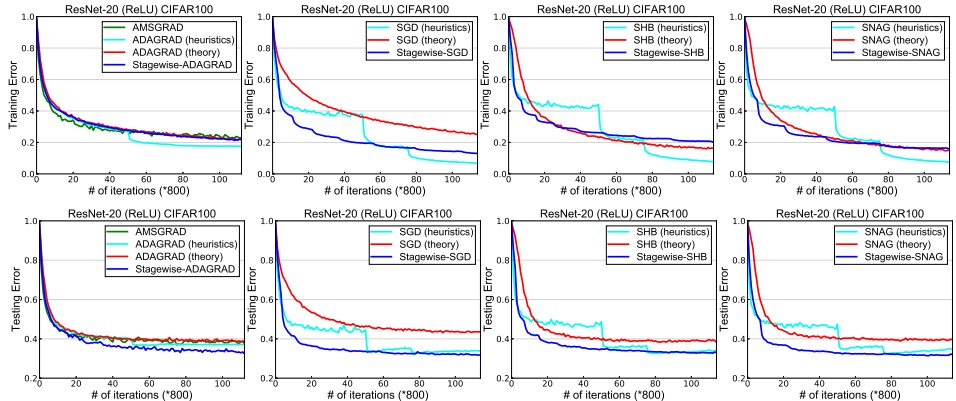

Figure 5: Comparison of Training Error (Top) and Testing Error (bottom) on CIFAR-100 with Regularization. The regularization parameter is set $5e - 4$.

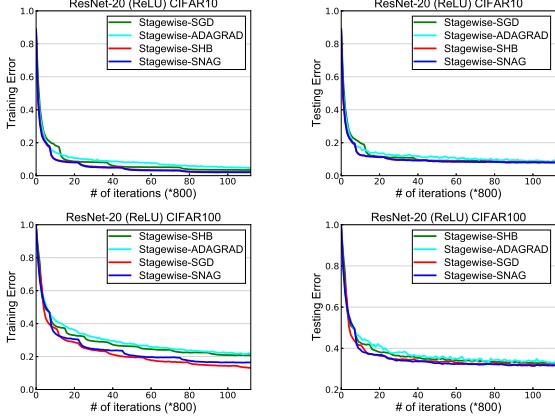

Figure 6: Comparison of different stagewise algorithms in terms of Training Error and Testing Error on CIFAR-10 (top) and CIFAR-100 (bottom) with regularization. The regularization parameter is set $5e - 4$.

where the second inequality uses the strong convexity of $f_s(\mathbf{x})$, whose strong convexity parameter is $\gamma^{-1} - \mu$. Next, we bound $f_s(\mathbf{x}_{s-1}) - f_s(\mathbf{z}_s)$ given that $\mathbf{x}_{s-1}$ is fixed. According to the definition

of $f_s(\cdot)$, we have

$$f_s(\mathbf{x}_{s-1}) - f_s(\mathbf{z}_s) = \phi(\mathbf{x}_{s-1}) - \phi(\mathbf{z}_s) - \frac{1}{2\gamma}\|\mathbf{z}_s - \mathbf{x}_{s-1}\|^2$$

$$= \phi(\mathbf{x}_{s-1}) - \phi(\mathbf{x}_s) + \phi(\mathbf{x}_s) - \phi(\mathbf{z}_s) - \frac{1}{2\gamma}\|\mathbf{z}_s - \mathbf{x}_{s-1}\|^2$$

$$= [\phi(\mathbf{x}_{s-1}) - \phi(\mathbf{x}_s)] + \left[f_s(\mathbf{x}_s) - f_s(\mathbf{z}_s) + \frac{1}{2\gamma}\|\mathbf{z}_s - \mathbf{x}_{s-1}\|^2 - \frac{1}{2\gamma}\|\mathbf{x}_s - \mathbf{x}_{s-1}\|^2\right] - \frac{1}{2\gamma}\|\mathbf{z}_s - \mathbf{x}_{s-1}\|^2$$

$$\leq [\phi(\mathbf{x}_{s-1}) - \phi(\mathbf{x}_s)] + [f_s(\mathbf{x}_s) - f_s(\mathbf{z}_s)].$$

Taking expectation over randomness in the $s$-th stage on both sides, we have

$$f_s(\mathbf{x}_{s-1}) - f_s(\mathbf{z}_s) \leq \mathrm{E}_s[\phi(\mathbf{x}_{s-1}) - \phi(\mathbf{x}_s)] + \mathrm{E}_s[f_s(\mathbf{x}_s) - f_s(\mathbf{z}_s)]$$

$$\leq \mathrm{E}[\phi(\mathbf{x}_{s-1}) - \phi(\mathbf{x}_s)] + \varepsilon_1(\eta_s, T_s, \Theta)\|\mathbf{x}_{s-1} - \mathbf{z}_s\|_2^2 + \varepsilon_2(\eta_s, T_s, \Theta)(f_s(\mathbf{x}_{s-1}) - f_s(\mathbf{z}_s)) + \varepsilon_3(\eta_s, T_s, \Theta).$$

Thus,

$$(1 - \varepsilon_2(\eta_s, T_s, \Theta))(f_s(\mathbf{x}_{s-1}) - f_s(\mathbf{z}_s)) \leq \mathrm{E}[\phi(\mathbf{x}_{s-1}) - \phi(\mathbf{x}_s)] + \varepsilon_1(\eta_s, T_s, \Theta)\|\mathbf{x}_{s-1} - \mathbf{z}_s\|_2^2 + \varepsilon_3(\eta_s, T_s, \Theta).$$

Assuming that $\varepsilon_2(\eta_s, T_s, \Theta) \leq 1/2$, we have

$$\varepsilon_2(\eta_s, T_s, \Theta)(f_s(\mathbf{x}_{s-1}) - f_s(\mathbf{z}_s)) \leq \mathrm{E}_s[\phi(\mathbf{x}_{s-1}) - \phi(\mathbf{x}_s)] + \varepsilon_1(\eta_s, T_s, \Theta)\|\mathbf{x}_{s-1} - \mathbf{z}_s\|_2^2 + \varepsilon_3(\eta_s, T_s, \Theta).$$

Plugging this upper bound into (10), we have

$$\mathrm{E}_s\left[\frac{(1 - \alpha_s)}{2\gamma}\|\mathbf{x}_{s-1} - \mathbf{z}_s\|^2\right] \leq \mathrm{E}_s\left[\phi(\mathbf{x}_{s-1}) - \phi(\mathbf{x}_s)\right]$$

$$+ \mathrm{E}_s\left[\frac{\alpha_s^{-1} - \gamma\mu}{(1 - \gamma\mu)}\{2\varepsilon_1(\eta_s, T_s, \Theta)\|\mathbf{x}_{s-1} - \mathbf{z}_s\|^2 + \phi(\mathbf{x}_{s-1}) - \phi(\mathbf{x}_s) + 2\varepsilon_3(\eta_s, T_s, \Theta)\}\right] \quad (11)$$

By setting $\alpha_s = 1/2, \gamma = 1/(2\mu)$ and assuming $\varepsilon_1(\eta_s, T_s, \Theta) \leq 1/(48\gamma)$, we have

$$\mathrm{E}_s\left[\frac{1}{8\gamma}\|\mathbf{x}_{s-1} - \mathbf{z}_s\|^2\right] \leq 4\mathrm{E}_s\left[\phi(\mathbf{x}_{s-1}) - \phi(\mathbf{x}_s)\right] + 6\varepsilon_3(\eta_s, T_s, \Theta)\}$$

Define $w_s = s^\alpha$. Multiplying both sides by $w_s$, we have that

$$w_s\gamma\mathrm{E}_s[\|\nabla\phi_\gamma(\mathbf{x}_{s-1})\|^2] \leq \mathrm{E}_s\left[32w_s\Delta_s + 48\varepsilon_3(\eta_s, T_s, \Theta)w_s\right]$$

By summing over $s = 1, \ldots, S + 1$, we have

$$\sum_{s=1}^{S+1} w_s\mathrm{E}[\|\nabla\phi_\gamma(\mathbf{x}_{s-1})\|^2] \leq \mathrm{E}\left[\frac{32}{\gamma}\sum_{s=1}^{S+1} w_s\Delta_s + \frac{48}{\gamma}\sum_{s=1}^{S+1} w_s\varepsilon_3(\eta_s, T_s, \Theta)\right]$$

Taking the expectation w.r.t. $\tau \in \{0, \ldots, S\}$, we have that

$$\mathrm{E}[\|\nabla\phi_\gamma(\mathbf{x}_\tau)\|^2]] \leq \mathrm{E}\left[\frac{32\sum_{s=1}^{S+1} w_s\Delta_s}{\gamma\sum_{s=1}^{S+1} w_s} + \frac{48\sum_{s=1}^{S+1} w_s\varepsilon_3(\eta_s, T_s, \Theta))}{\gamma\sum_{s=1}^{S+1} w_s}\right]$$

For the first term on the R.H.S, we have that

$$\sum_{s=1}^{S+1} w_s\Delta_s = \sum_{s=1}^{S+1} w_s(\phi(\mathbf{x}_{s-1}) - \phi(\mathbf{x}_s)) = \sum_{s=1}^{S+1}(w_{s-1}\phi(\mathbf{x}_{s-1}) - w_s\phi(\mathbf{x}_s)) + \sum_{s=1}^{S+1}(w_s - w_{s-1})\phi(\mathbf{x}_{s-1})$$

$$= w_0\phi(\mathbf{x}_0) - w_{S+1}\phi(\mathbf{x}_{S+1}) + \sum_{s=1}^{S+1}(w_s - w_{s-1})\phi(\mathbf{x}_{s-1})$$

$$= \sum_{s=1}^{S+1}(w_s - w_{s-1})(\phi(\mathbf{x}_{s-1}) - \phi(\mathbf{x}_{S+1})) \leq \Delta_\phi\sum_{s=1}^{S+1}(w_s - w_{s-1}) = \Delta_\phi w_{S+1}$$

Then,

$$E[\|\nabla\phi_\gamma(\mathbf{x}_\tau)\|^2] \le \frac{32\Delta_\phi w_{S+1}}{\gamma\sum_{s=1}^{S+1} w_s} + \frac{48\sum_{s=1}^{S+1} w_s\varepsilon_3(\eta_s, T_s, \Theta)}{\gamma\sum_{s=1}^{S+1} w_s}$$

The standard calculus tells that

$$\sum_{s=1}^{S} s^\alpha \ge \int_0^S x^\alpha dx = \frac{1}{\alpha+1}S^{\alpha+1}$$

$$\sum_{s=1}^{S} s^{\alpha-1} \le SS^{\alpha-1} = S^\alpha, \forall\alpha\ge 1, \quad \sum_{s=1}^{S} s^{\alpha-1} \le \int_0^S x^{\alpha-1}dx = \frac{S^\alpha}{\alpha}, \forall 0 < \alpha < 1$$

Combining these facts and the assumption $\varepsilon_3(\eta_s, T_s, \Theta) \le c/s$, we have that

$$E[\|\nabla\phi_\gamma(\mathbf{x}_\tau)\|^2] \le \begin{cases} \frac{32\Delta_\phi(\alpha+1)}{\gamma(S+1)} + \frac{48c(\alpha+1)}{\gamma(S+1)} & \alpha \ge 1 \\[2mm] \frac{32\Delta_\phi(\alpha+1)}{\gamma(S+1)} + \frac{48c(\alpha+1)}{\gamma(S+1)\alpha} & 0 < \alpha < 1 \end{cases}$$

In order to have $E[\|\nabla\phi_\gamma(\mathbf{x}_\tau)\|^2] \le \epsilon^2$, we can set $S = O(1/\epsilon^2)$. The total number of iterations is

$$\sum_{s=1}^{S} T_s \le \sum_{s=1}^{S} 12\gamma s \le 6\gamma S(S+1) = O(1/\epsilon^4)$$

$\square$

## C  PROOF OF THEOREM 2

*Proof.* The proof is almost a duplicate to that of Theorem 1. Define $w_s = s^\alpha$. We apply Lemma 1 to each call of SGD in stagewise SGD,

$$E[f_s(\mathbf{x}_s) - f_s(\mathbf{z}_s)] \le \underbrace{\frac{\|\mathbf{z}_s - \mathbf{x}_{s-1}\|^2}{2\eta_s T_s} + \frac{\eta_s\hat{G}^2}{2}}_{\mathcal{E}_s},$$

where $\hat{G}^2$ is the upper bound of $E[\|\mathbf{g}(\mathbf{x};\xi) + \gamma^{-1}(\mathbf{x} - \mathbf{x}_{s-1})\|^2]$, which exists and can be set to $2G^2 + 2\gamma^{-2}D^2$ due to the Assumption 1-(ii) and the bounded assumption of the domain. Then following the same analysis as that in the proof of Theorem 1, we have

$$\begin{aligned} E_s\left[\frac{(1-\alpha_s)}{2\gamma}\|\mathbf{x}_{s-1} - \mathbf{z}_s\|^2\right] &\le E_s\left[\phi(\mathbf{x}_{s-1}) - \phi(\mathbf{x}_s) + \frac{(\alpha_s^{-1}-1)}{2\gamma}\|\mathbf{x}_s - \mathbf{z}_s\|^2 + \mathcal{E}_s\right] \\ &\le E\left[\phi(\mathbf{x}_{s-1}) - \phi(\mathbf{x}_s) + \frac{(\alpha_s^{-1}-1)}{\gamma(\gamma^{-1}-\mu)}(f_s(\mathbf{x}_s) - f_s(\mathbf{z}_s)) + \mathcal{E}_s\right] \\ &\le E\left[\phi(\mathbf{x}_{s-1}) - \phi(\mathbf{x}_s) + \frac{\alpha_s^{-1}-\gamma\mu}{(1-\gamma\mu)}\mathcal{E}_s\right] \end{aligned} \tag{12}$$

Combining the above inequalities, we have that

$$\left((1-\alpha_s)\gamma - \frac{\gamma^2(\alpha_s^{-1}-\mu\gamma)}{(1-\mu\gamma)\eta_s T_s}\right)E_s[\|\nabla\phi_\gamma(\mathbf{x}_{s-1})\|^2] \le E_s\left[2\Delta_s + \frac{(\alpha_s^{-1}-\mu\gamma)\eta_s\hat{G}^2}{(1-\mu\gamma)}\right]$$

Multiplying both sides by $w_s$, we have that

$$w_s\left((1-\alpha_s)\gamma - \frac{\gamma^2(\alpha_s^{-1}-\mu\gamma)}{(1-\mu\gamma)\eta_s T_s}\right)E_s[\|\nabla\phi_\gamma(\mathbf{x}_{s-1})\|^2] \le E_s\left[2w_s\Delta_s + \frac{(\alpha_s^{-1}-\mu\gamma)w_s\eta_s\hat{G}^2}{(1-\mu\gamma)}\right]$$

By setting $\alpha_s = 1/2$ and $\gamma = 1/(2\mu)$, $T_s\eta_s \ge 12\gamma$, we have

$$\frac{1}{4}w_s\gamma E_s[\|\nabla\phi_\gamma(\mathbf{x}_{s-1})\|^2] \le E_s[2w_s\Delta_s + 3w_s\eta_s\hat{G}^2]$$

By summing over $s = 1, \ldots, S + 1$, we have

$$\sum_{s=1}^{S+1} w_s \mathrm{E}[\|\nabla\phi_\gamma(\mathbf{x}_{s-1})\|^2] \leq \mathrm{E}\left[16\mu \sum_{s=1}^{S+1} w_s \Delta_s + 24\mu \sum_{s=1}^{S+1} w_s \eta_s \hat{G}^2\right]$$

Taking the expectation w.r.t. $\tau \in \{0, \ldots, S\}$, we have that

$$\mathrm{E}[\|\nabla\phi_\gamma(\mathbf{x}_\tau)\|^2]] \leq \mathrm{E}\left[\frac{16\mu \sum_{s=1}^{S+1} w_s \Delta_s}{\sum_{s=1}^{S+1} w_s} + \frac{24\mu \sum_{s=1}^{S+1} w_s \eta_s \hat{G}^2}{\sum_{s=1}^{S+1} w_s}\right]$$

By similar analysis, we have that

$$\mathrm{E}[\|\nabla\phi_\gamma(\mathbf{x}_\tau)\|^2] \leq \begin{cases} \frac{16\mu\Delta_\phi(\alpha+1)}{S+1} + \frac{24\eta_0\mu\hat{G}^2(\alpha+1)}{S+1} & \alpha \geq 1 \\[2mm] \frac{16\mu\Delta_\phi(\alpha+1)}{S+1} + \frac{24\eta_0\mu\hat{G}^2(\alpha+1)}{(S+1)\alpha} & 0 < \alpha < 1 \end{cases}$$

In order to have $\mathrm{E}[\|\nabla\phi_\gamma(\mathbf{x}_\tau)\|^2] \leq \epsilon^2$, we can set $S = O(1/\epsilon^2)$. The total number of iterations is

$$\sum_{s=1}^{S} T_s \leq \sum_{s=1}^{S} 12\gamma s \leq 6\gamma S(S+1) = O(1/\epsilon^4)$$

$\square$

## D    PROOF OF THEOREM 4

We need the following lemma for the convergence bound of stochastic momentum methods for a strongly convex problem, whose proof is postponed to Section F.

**Lemma 3.** *For Algorithm 4, assume $f(\mathbf{x}) = \phi(\mathbf{x}) + \frac{1}{2\gamma}\|\mathbf{x} - \mathbf{x}_0\|^2$ is a $\lambda$-strongly convex function, $\mathbf{g}_t = \mathbf{g}(\mathbf{x}_t; \xi) + \frac{1}{\gamma}(\mathbf{x}_t - \mathbf{x}_0)$ where $\mathbf{g}(\mathbf{x}; \xi) \in \partial_F\phi(\mathbf{x}_t)$ such that $\mathrm{E}[\|\mathbf{g}(\mathbf{x}; \xi)\|^2] \leq G^2$, and $\eta \leq (1-\beta)\gamma^2\lambda/(8\rho\beta + 4)$, then we have that*

$$\mathrm{E}[f(\hat{\mathbf{x}}_T) - f(\mathbf{x}_*)] \leq$$
$$\frac{(1-\beta)\|\mathbf{x}_0 - \mathbf{x}_*\|^2}{2\eta(T+1)} + \frac{\beta(f(\mathbf{x}_0) - f(\mathbf{x}_*))}{(1-\beta)(T+1)} + \frac{2\eta G^2(2\rho\beta + 1)}{1-\beta} + \frac{4\rho\beta + 4}{(1-\beta)}\frac{\eta}{\gamma^2}\|\mathbf{x}_0 - \mathbf{x}_*\|^2 \quad (13)$$

*where $\hat{\mathbf{x}}_T = \sum_{t=0}^{T} \mathbf{x}_t/(1 + T)$ and $\mathbf{x}_* \in \arg\min_{\mathbf{x}\in\mathbb{R}^d} f(\mathbf{x})$.*

**Remark:** It is notable that in the above result, we do not use the bounded domain assumption since we consider $\Omega = \mathbb{R}^d$ for the unified momentum methods in this subsection. The key to get rid of bounded domain assumption is by exploring the strong convexity of $f(\mathbf{x}) = \phi(\mathbf{x}) + \frac{1}{2\gamma}\|\mathbf{x} - \mathbf{x}_0\|^2$.

*Proof.* of Theorem 4 According to the definition of $\mathbf{z}_s$ in (8) and Lemma 3, we have that

$$\mathrm{E}_s\left[\phi(\mathbf{x}_s) + \frac{1}{2\gamma}\|\mathbf{x}_s - \mathbf{x}_{s-1}\|^2\right]$$
$$\leq f_s(\mathbf{z}_s) + \underbrace{\frac{\beta(f_s(\mathbf{x}_{s-1}) - f_s(\mathbf{z}_s))}{(1-\beta)(T_s + 1)} + \frac{(1-\beta)\|\mathbf{x}_{s-1} - \mathbf{z}_s\|^2}{2\eta_s(T_s + 1)} + \frac{2\eta_s G^2(2\rho\beta + 1)}{1-\beta} + \frac{1}{24\gamma}\|\mathbf{x}_{s-1} - \mathbf{z}_s\|^2}_{\mathcal{E}_s}$$

$$\leq \phi(\mathbf{x}_{s-1}) + \mathcal{E}_s,$$

where the last inequality uses the value of $\eta_s = (1-\beta)\gamma/(96s(\rho\beta+1)) \leq (1-\beta)\gamma/(96(\rho\beta+1))$, which also satisfies the condition in Lemma 3 by noting that $\lambda = \gamma^{-1} - \mu = 1/(2\gamma)$. Similar to the proof of Theorem 1, we have

$$\frac{(1-\alpha_s)}{2\gamma}\|\mathbf{x}_{s-1} - \mathbf{z}_s\|^2 \leq \mathrm{E}_s[\phi(\mathbf{x}_{s-1}) - \phi(\mathbf{x}_s)] + \frac{\alpha_s^{-1} - \gamma\mu}{(1-\gamma\mu)}\mathcal{E}_s \quad (14)$$

Plugging the expression of $\mathcal{E}_s$ and rearranging above inequality, we have that

$$\left((1-\alpha_s)\gamma - \frac{\gamma^2(\alpha_s^{-1}-\mu\gamma)(1-\beta)}{(1-\mu\gamma)\eta_s(T_s+1)} - \frac{\alpha_s^{-1}-\gamma\mu}{(1-\gamma\mu)}\frac{\gamma}{24}\right)\|\nabla\phi_\gamma(\mathbf{x}_{s-1})\|^2$$

$$\leq 2\mathrm{E}_s[\Delta_s] + \frac{2(\alpha_s^{-1}-\mu\gamma)}{(1-\mu\gamma)}\left[\frac{\beta(f_s(\mathbf{x}_{s-1})-f_s(\mathbf{z}_s))}{(1-\beta)(T_s+1)} + \frac{2\eta_s\hat{G}^2(2\rho\beta+1)}{1-\beta}\right]$$

The definition of $f_s$ gives that

$$f_s(\mathbf{x}_{s-1}) - f_s(\mathbf{z}_s) = \phi(\mathbf{x}_{s-1}) - \phi(\mathbf{z}_s) - \frac{1}{2\gamma}\|\mathbf{z}_s - \mathbf{x}_{s-1}\|^2$$

On the other hand, the $\mu$-weakly convexity of $\phi$ gives that

$$\phi(\mathbf{z}_s) \geq \phi(\mathbf{x}_{s-1}) + \langle \mathbf{g}(\mathbf{x}_{s-1}), \mathbf{z}_s - \mathbf{x}_{s-1}\rangle - \frac{\mu}{2}\|\mathbf{z}_s - \mathbf{x}_{s-1}\|^2,$$

where $\mathbf{g}(\mathbf{x}_{s-1}) \in \partial_F\phi(\mathbf{x}_{s-1})$. Combing these two inequalities we have that

$$f_s(\mathbf{x}_{s-1}) - f_s(\mathbf{z}_s) \leq \langle \mathbf{g}(\mathbf{x}_{s-1}), \mathbf{x}_{s-1} - \mathbf{z}_s\rangle - \frac{\mu}{2}\|\mathbf{z}_s - \mathbf{x}_{s-1}\|^2$$

$$\leq \frac{G^2}{2\mu} + \frac{\mu-\mu}{2}\|\mathbf{z}_s - \mathbf{x}_{s-1}\|^2 = \frac{G^2}{2\mu}$$

where the second inequality follows from Jensen's inequality for $\|\cdot\|$ and Young's inequality. Combining above inequalities and multiplying both side by $w_s$, we have that

$$w_s\left((1-\alpha_s)\gamma - \frac{\gamma^2(\alpha_s^{-1}-\mu\gamma)(1-\beta)}{(1-\mu\gamma)\eta_s(T_s+1)} - \frac{\alpha_s^{-1}-\gamma\mu}{(1-\gamma\mu)}\frac{\gamma}{24}\right)\|\nabla\phi_\gamma(\mathbf{x}_{s-1})\|^2$$

$$\leq 2w_s\mathrm{E}_s[\Delta_s] + \frac{2w_s(\alpha_s^{-1}-\mu\gamma)}{(1-\mu\gamma)}\left[\frac{\beta G^2}{2\mu(1-\beta)(T_s+1)} + \frac{2\eta_s G^2(2\rho\beta+1)}{1-\beta}\right] \qquad (15)$$

By setting $\alpha_s = 1/2$, $\eta_s(T_s+1) \geq 24(1-\beta)\gamma$, we have that

$$\frac{w_s\gamma}{4}\|\nabla\phi_\gamma(\mathbf{x}_{s-1})\|^2 \leq 2w_s\mathrm{E}_s[\Delta_s] + \frac{w_s\eta_s\beta G^2}{4(1-\beta)^2} + \frac{12w_s\eta_s G^2(2\rho\beta+1)}{1-\beta}$$

Summing over $s = 1,\ldots,S+1$ and rearranging, we have

$$\sum_{s=1}^{S+1} w_s\|\nabla\phi_\gamma(\mathbf{x}_{s-1})\|^2 = \mathrm{E}\left[\sum_{s=1}^{S+1}\frac{8}{\gamma}w_s\Delta_s + \frac{w_s\eta_s G^2(\beta + 48(2\rho\beta+1)(1-\beta))}{\gamma(1-\beta)^2}\right]$$

Following similar analysis as in the proof of Theorem 2, we can finish the proof. $\qquad\square$

## E  PROOF OF THEOREM 3

*Proof.* Applying Lemma 2 with $T_s \geq M_s\max\{\frac{\hat{G}+\max_i\|g_{1:T_s,i}^s\|}{2c}, c\sum_{i=1}^d\|g_{1:T_s,i}^s\|\}$ $M_s > 0$, and the fact that $\phi(\mathbf{x}_{s-1}) \geq \phi(\mathbf{z}_s) + \frac{1}{2\gamma}\|\mathbf{x}_{s-1}-\mathbf{z}_s\|^2$ in $s$th stage, we have that

$$\mathrm{E}_s\left[\phi(\mathbf{x}_s) + \frac{1}{2\gamma_s}\|\mathbf{x}_s - \mathbf{x}_{s-1}\|^2\right] \leq f_s(\mathbf{z}_s) + \underbrace{\frac{c}{M_s\eta_s}\|\mathbf{x}_{s-1}-\mathbf{z}_s\|^2 + \frac{\eta_s}{M_s c}}_{\mathcal{E}_s}$$

$$\leq \phi(\mathbf{x}_s) + \mathcal{E}_s$$

According to (14), we have that

$$\frac{(1-\alpha_s)}{2\gamma}\mathrm{E}_s[\|\mathbf{x}_{s-1}-\mathbf{z}_s\|^2] \leq \phi(\mathbf{x}_{s-1}) - \phi(\mathbf{x}_s) + \frac{(\alpha_s^{-1}-1)}{2\gamma}\|\mathbf{x}_s - \mathbf{z}_s\|^2 + \mathcal{E}_s$$

$$\leq \phi(\mathbf{x}_{s-1}) - \phi(\mathbf{x}_s) + \frac{\alpha_s^{-1}-\gamma\mu}{(1-\gamma\mu)}\left(\frac{c}{M_s\eta_s}\|\mathbf{x}_{s-1}-\mathbf{z}_s\|^2 + \frac{\eta_s}{M_s c}\right)$$

Rearranging above inequality then multiplying both side by $w_s$, we have that

$$w_s\left((1-\alpha_s)\gamma - \frac{2\gamma^2 c(\alpha_s^{-1} - \mu\gamma)}{(1-\mu\gamma)M_s\eta_s}\right)\|\nabla\phi_\gamma(\mathbf{x}_{s-1})\|^2$$

$$\leq 2w_s\mathrm{E}_s[\Delta_s] + \frac{2w_s\eta_s(\alpha_s^{-1} - \mu\gamma)}{cM_s(1-\mu\gamma)}$$

By using $M_s\eta_s \geq 24\gamma c$ and summing over $s = 1, \ldots, S+1$, we have that

$$\sum_{s=1}^{S+1} w_s\|\nabla\phi_\gamma(\mathbf{x}_{s-1})\|^2 \leq \mathrm{E}\left[\sum_{s=1}^{S+1} \frac{8w_s\Delta_s}{\gamma} + \frac{w_s\eta_s^2}{c^2\gamma^2}\right]$$

By the definition of $\tau$ in the theorem, taking expectation of $\|\nabla\phi_\gamma(\mathbf{x}_\tau)\|^2$ w.r.t. $\tau \in \{0, \ldots, S\}$ we have that

$$\mathrm{E}[\|\nabla\phi_\gamma(\mathbf{x}_\tau)\|^2] = \mathrm{E}\left[\frac{8}{\gamma}\sum_{s=1}^{S+1}\frac{w_s\Delta_s}{\sum_{i=1}^{S+1}w_i}\right] + \frac{\eta_0^2}{c^2\gamma^2}\sum_{s=1}^{S+1}\frac{s^{\alpha-1}}{\sum_{i=1}^{S+1}w_i}$$

$$\leq \frac{8\Delta_\phi(\alpha+1)}{\gamma(S+1)} + \frac{\eta_0^2(\alpha+1)}{c^2\gamma^2(S+1)\alpha^{\mathbb{I}(\alpha<1)}},$$

where $\mathbb{I}(\alpha < 1)$ is 1 if $\alpha < 1$ and 0 otherwise.

$\square$

# F   PROOF OF LEMMA 3

*Proof.* Following the analysis in Yang et al. (2016), we directly have the following inequality,

$$\mathrm{E}[\|\mathbf{x}_{k+1} + \mathbf{p}_{k+1} - \mathbf{x}_*\|^2] =$$

$$= \mathrm{E}[\|\mathbf{x}_k + \mathbf{p}_k - \mathbf{x}_*\|^2] - \frac{2\eta}{1-\beta}\mathrm{E}[(\mathbf{x}_k - \mathbf{x}_*)^\top\partial f(\mathbf{x}_k)] - \frac{2\eta\beta}{(1-\beta)^2}\mathrm{E}[(\mathbf{x}_k - \mathbf{x}_{k-1})^\top\partial f(\mathbf{x}_k)]$$

$$- \frac{2\rho\eta^2\beta}{(1-\beta)^2}\mathrm{E}[\mathbf{g}_{k-1}^\top\partial f(\mathbf{x}_k)] + \left(\frac{\eta}{1-\beta}\right)^2\mathrm{E}[\|\mathbf{g}_k\|^2]$$

We also note that

$$f(\mathbf{x}_k) - f(\mathbf{x}_*) \leq (\mathbf{x}_k - \mathbf{x}_*)^\top\partial f(\mathbf{x}_k) - \frac{\lambda}{2}\|\mathbf{x}_k - \mathbf{x}_*\|^2$$

$$f(\mathbf{x}_k) - f(\mathbf{x}_{k-1}) \leq (\mathbf{x}_k - \mathbf{x}_{k-1})^\top\partial f(\mathbf{x}_k) - \frac{\lambda}{2}\|\mathbf{x}_k - \mathbf{x}_{k-1}\|^2$$

$$- \mathrm{E}[\mathbf{g}_{k-1}^\top\partial f(\mathbf{x}_k)] \leq \frac{\mathrm{E}[\|\mathbf{g}_{k-1}\|^2 + \|\partial f(\mathbf{x}_k)\|^2]}{2} \leq \frac{1}{\gamma^2}\|\mathbf{x}_{k-1} - \mathbf{x}_0\|^2 + \frac{1}{\gamma^2}\|\mathbf{x}_k - \mathbf{x}_0\|^2 + 2G^2$$

$$\mathrm{E}_k[\|\mathbf{g}_k\|^2] \leq \frac{2}{\gamma^2}\|\mathbf{x}_k - \mathbf{x}_0\|^2 + 2G^2$$

where the first two inequalities are due to the strong convexity of $f(\cdot)$ and the last three inequalities are due to the boundness assumption. Thus

$$\mathrm{E}[\|\mathbf{x}_{k+1} + \mathbf{p}_{k+1} - \mathbf{x}\|^2] \leq \mathrm{E}[\|\mathbf{x}_k + \mathbf{p}_k - \mathbf{x}\|^2] - \frac{2\eta}{1-\beta}\mathrm{E}[(f(\mathbf{x}_k) - f(\mathbf{x}))]$$

$$- \frac{2\eta\beta}{(1-\beta)^2}\mathrm{E}[(f(\mathbf{x}_k) - f(\mathbf{x}_{k-1}))] + \left(\frac{\eta}{1-\beta}\right)^2(2\rho\beta+1)4G^2$$

$$- \frac{\lambda\eta}{1-\beta}\|\mathbf{x}_k - \mathbf{x}_*\|^2 - \frac{\lambda\eta\beta}{(1-\beta)^2}\|\mathbf{x}_k - \mathbf{x}_{k-1}\|^2$$

$$+ \frac{2\rho\beta}{(1-\beta)^2}\frac{\eta^2}{\gamma^2}\|\mathbf{x}_{k-1} - \mathbf{x}_0\|^2 + \frac{2\rho\beta+2}{(1-\beta)^2}\frac{\eta^2}{\gamma^2}\|\mathbf{x}_k - \mathbf{x}_0\|^2$$

By summarizing the above inequality over $k = 0, \ldots, T$, we have

$$
\frac{2\eta}{1-\beta} \mathrm{E}\Big[ \sum_{k=0}^{T} (f(\mathbf{x}_k) - f(\mathbf{x}_*)) \Big] \le \mathrm{E}[\|\mathbf{x}_0 - \mathbf{x}_*\|^2] + \frac{2\eta\beta}{(1-\beta)^2} \mathrm{E}[f(\mathbf{x}_0) - f(\mathbf{x}_*)]
$$

$$
+ \left( \frac{\eta}{1-\beta} \right)^2 (2\rho\beta + 1) 4G^2 (T+1)
$$

$$
- \frac{\eta\lambda}{1-\beta} \sum_{k=0}^{T} \|\mathbf{x}_k - \mathbf{x}_*\|^2 + \frac{4\rho\beta}{(1-\beta)^2} \frac{\eta^2}{\gamma^2} \sum_{k=0}^{T} \|\mathbf{x}_{k-1} - \mathbf{x}_*\|^2 + \frac{4\rho\beta + 4}{(1-\beta)^2} \frac{\eta^2}{\gamma^2} \sum_{k=0}^{T} \|\mathbf{x}_k - \mathbf{x}_*\|^2
$$

$$
+ \frac{4\rho\beta + 4}{(1-\beta)^2} \frac{\eta^2}{\gamma^2} (T+1) \|\mathbf{x}_0 - \mathbf{x}_*\|^2
$$

When $\eta \le (1-\beta)\gamma^2 \lambda / (8\rho\beta + 4)$, we have

$$
\mathrm{E}\Big[ (f(\widehat{\mathbf{x}}_T) - f(\mathbf{x}_*)) \Big] \le \frac{(1-\beta)\|\mathbf{x}_0 - \mathbf{x}_*\|^2}{2\eta(T+1)} + \frac{\beta}{(1-\beta)} \frac{f(\mathbf{x}_0) - f(\mathbf{x}_*)}{T+1} + \frac{\eta}{1-\beta}(2\rho\beta + 1)2G^2
$$

$$
+ \frac{4\rho\beta + 4}{(1-\beta)} \frac{\eta}{\gamma^2} \|\mathbf{x}_0 - \mathbf{x}_*\|^2
$$

$$
\square
$$

## G  PROOF OF LEMMA 2

The proof is almost a duplicate of the proof of Proposition 1 in Chen et al. (2018b). For completeness, we present a proof here.

*Proof.* Let $\psi_0(\mathbf{x}) = 0$ and $\|\mathbf{x}\|_H = \sqrt{\mathbf{x}^\top H \mathbf{x}}$. First, we can see that $\psi_{t+1}(\mathbf{x}) \ge \psi_t(\mathbf{x})$ for any $t \ge 0$. Define $\zeta_t = \sum_{\tau=1}^{t} \mathbf{g}_t$ and $\Delta_\tau = (\partial F(\mathbf{x}_t) - \mathbf{g}_t)^\top (\mathbf{x}_t - \mathbf{x})$. Let $\psi_t^*$ be defined by

$$
\psi_t^*(g) = \sup_{\mathbf{x} \in \Omega} g^\top \mathbf{x} - \frac{1}{\eta} \psi_t(\mathbf{x})
$$

Taking the summation of objective gap in all iterations, we have

$$
\sum_{t=1}^{T} (f(\mathbf{x}_t) - f(\mathbf{x})) \le \sum_{t=1}^{T} \partial f(\mathbf{x}_t)^\top (\mathbf{x}_t - \mathbf{x}) = \sum_{t=1}^{T} \mathbf{g}_t^\top (\mathbf{x}_t - \mathbf{x}) + \sum_{t=1}^{T} \Delta_t
$$

$$
= \sum_{t=1}^{T} \mathbf{g}_t^\top \mathbf{x}_t - \sum_{t=1}^{T} \mathbf{g}_t^\top \mathbf{x} - \frac{1}{\eta} \psi_T(\mathbf{x}) + \frac{1}{\eta} \psi_T(\mathbf{x}) + \sum_{t=1}^{T} \Delta_t
$$

$$
\le \frac{1}{\eta} \psi_T(\mathbf{x}) + \sum_{t=1}^{T} \mathbf{g}_t^\top \mathbf{x}_t + \sum_{t=1}^{T} \Delta_t + \sup_{\mathbf{x} \in \Omega} \left\{ -\sum_{t=1}^{T} \mathbf{g}_t^\top \mathbf{x} - \frac{1}{\eta} \psi_T(\mathbf{x}) \right\}
$$

$$
= \frac{1}{\eta} \psi_T(\mathbf{x}) + \sum_{t=1}^{T} \mathbf{g}_t^\top \mathbf{x}_t + \psi_T^*(-\zeta_T) + \sum_{t=1}^{T} \Delta_t
$$

Note that

$$
\psi_T^*(-\zeta_T) = -\sum_{t=1}^{T} \mathbf{g}_t^\top \mathbf{x}_{T+1} - \frac{1}{\eta} \psi_T(\mathbf{x}_{T+1}) \le -\sum_{t=1}^{T} \mathbf{g}_t^\top \mathbf{x}_{T+1} - \frac{1}{\eta} \psi_{T-1}(\mathbf{x}_{T+1})
$$

$$
\le \sup_{\mathbf{x} \in \Omega} -\zeta_T^\top \mathbf{x} - \frac{1}{\eta} \psi_{T-1}(\mathbf{x}) = \psi_{T-1}^*(-\zeta_T)
$$

$$
\le \psi_{T-1}^*(-\zeta_{T-1}) - \mathbf{g}_T^\top \nabla \psi_{T-1}^*(-\zeta_{T-1}) + \frac{\eta}{2} \|\mathbf{g}_T\|_{\psi_{T-1}^*}^2
$$

where the last inequality uses the fact that $\psi_t(\mathbf{x})$ is 1-strongly convex w.r.t $\|\cdot\|_{\psi_t} = \|\cdot\|_{H_t}$ and consequentially $\psi_t^*(\mathbf{x})$ is $\eta$-smooth wr.t. $\|\cdot\|_{\psi_t^*} = \|\cdot\|_{H_t^{-1}}$. Thus, we have

$$\sum_{t=1}^{T} \mathbf{g}_t^\top \mathbf{x}_t + \psi_T^*(-\zeta_T) \le \sum_{t=1}^{T} \mathbf{g}_t^\top \mathbf{x}_t + \psi_{T-1}^*(-\zeta_{T-1}) - \mathbf{g}_T^\top \nabla \psi_{T-1}^*(-\zeta_{T-1}) + \frac{\eta}{2} \|\mathbf{g}_T\|_{\psi_{T-1}^*}^2$$

$$= \sum_{t=1}^{T-1} \mathbf{g}_t^\top \mathbf{x}_t + \psi_{T-1}^*(-\zeta_{T-1}) + \frac{\eta}{2} \|\mathbf{g}_T\|_{\psi_{T-1}^*}^2$$

By repeating this process, we have

$$\sum_{t=1}^{T} \mathbf{g}_t^\top \mathbf{x}_t + \psi_T^*(-\zeta_T) \le \psi_0^*(-\zeta_0) + \frac{\eta}{2} \sum_{t=1}^{T} \|\mathbf{g}_t\|_{\psi_{t-1}^*}^2 = \frac{\eta}{2} \sum_{t=1}^{T} \|\mathbf{g}_t\|_{\psi_{t-1}^*}^2$$

Then

$$\sum_{t=1}^{T} (f(\mathbf{x}_t) - f(\mathbf{x})) \le \frac{1}{\eta} \psi_T(\mathbf{x}) + \frac{\eta}{2} \sum_{t=1}^{T} \|\mathbf{g}_t\|_{\psi_{t-1}^*}^2 + \sum_{t=1}^{T} \Delta_t \tag{16}$$

Following the analysis in Duchi et al. (2011), we have

$$\sum_{t=1}^{T} \|\mathbf{g}_t\|_{\psi_{t-1}^*}^2 \le 2 \sum_{i=1}^{d} \|\mathbf{g}_{1:T,i}\|_2$$

Thus

$$\sum_{t=1}^{T} (f(\mathbf{x}_t) - f(\mathbf{x})) \le \frac{G\|\mathbf{x} - \mathbf{x}_1\|_2^2}{2\eta} + \frac{(\mathbf{x} - \mathbf{x}_1)^\top \mathrm{diag}(s_T)(\mathbf{x} - \mathbf{x}_1)}{2\eta} + \eta \sum_{i=1}^{d} \|\mathbf{g}_{1:T,i}\|_2 + \sum_{t=1}^{T} \Delta_t$$

$$\le \frac{G + \max_i \|\mathbf{g}_{1:T,i}\|_2}{2\eta} \|\mathbf{x} - \mathbf{x}_1\|_2^2 + \eta \sum_{i=1}^{d} \|\mathbf{g}_{1:T,i}\|_2 + \sum_{t=1}^{T} \Delta_t$$

Now by the value of $T \ge M \max\{ \frac{(G + \max_i \|g_{1:T,i})\|}{2c}, c \sum_{i=1}^{d} \|g_{1:T,i}\| \}$, we have

$$\frac{(G + \max_i \|g_{1:T,i}\|_2)}{2\eta T} \le \frac{c}{\eta M},$$

$$\frac{\eta \sum_{i=1}^{d} \|g_{1:T,i}\|_2}{T} \le \frac{\eta}{Mc}$$

Dividing by $T$ on both sides and setting $\mathbf{x} = \mathbf{x}_*$, following the inequality (3) and the convexity of $f(\mathbf{x})$ we have

$$f(\widehat{\mathbf{x}}) - f_* \le \frac{c}{M\eta} \|\mathbf{x}_0 - \mathbf{x}_*\|^2 + \frac{\eta}{Mc} + \frac{1}{T} \sum_{t=1}^{T} \Delta_t$$

Let $\{\mathcal{F}_t\}$ be the filtration associated with Algorithm 1 in the paper. Noticing that $T$ is a random variable with respect to $\{\mathcal{F}_t\}$, we cannot get rid of the last term directly. Define the Sequence $\{X_t\}_{t \in \mathbb{N}_+}$ as

$$X_t = \frac{1}{t} \sum_{i=1}^{t} \Delta_i = \frac{1}{t} \sum_{i=1}^{t} \langle \mathbf{g}_i - \mathrm{E}[\mathbf{g}_i], \mathbf{x}_i - \mathbf{x}_* \rangle \tag{17}$$

where $\mathrm{E}[\mathbf{g}_i] \in \partial f(\mathbf{x}_i)$. Since $\mathrm{E}\left[\mathbf{g}_{t+1} - \mathrm{E}[\mathbf{g}_{t+1}]\right] = 0$ and $\mathbf{x}_{t+1} = \arg\min_{\mathbf{x} \in \Omega} \eta \mathbf{x}^\top \left( \frac{1}{t} \sum_{\tau=1}^{t} \mathbf{g}_\tau \right) + \frac{1}{t} \psi_t(\mathbf{x})$, which is measurable with respect to $\mathbf{g}_1, \ldots, \mathbf{g}_t$ and $\mathbf{x}_1, \ldots, \mathbf{x}_t$, it is easy to see $\{\Delta_t\}_{t \in N}$ is a martingale difference sequence with respect to $\{\mathcal{F}_t\}$, e.g. $\mathrm{E}[\Delta_t | \mathcal{F}_{t-1}] = 0$. On the other hand, since $\|\mathbf{g}_t\|_2$ is upper bounded by $G$, following the statement of $T$ in the theorem, $T \le N = M^2 \max\{ \frac{(G+1)^2}{4}, d^2 G^2 \} < \infty$ always holds. Then following Lemma 1 in (Chen et al., 2018b) we have that $\mathrm{E}[X_T] = 0$.

Now taking the expectation we have that

$$\mathrm{E}[f(\widehat{\mathbf{x}}) - f_*] \le \frac{c}{M\eta} \|\mathbf{x}_0 - \mathbf{x}_*\|^2 + \frac{\eta c}{M}$$

Then we finish the proof. $\qquad \square$

## H    PROOF OF INEQUALITY (3)

Let us recall the definition of $\widehat{\mathbf{x}} = \operatorname{prox}_{\gamma f}(\mathbf{x})$.

$$\operatorname{prox}_{\gamma f}(\mathbf{x}) = \arg\min_{\mathbf{z}} f(\mathbf{z}) + \frac{1}{2\gamma}\|\mathbf{z} - \mathbf{x}\|^2 \tag{18}$$

First, note that when $f(\mathbf{z})$ is $\mu$-weakly convex and $\gamma < \mu^{-1}$, the above problem is strongly convex and $\widehat{\mathbf{x}} = \operatorname{prox}_{\gamma f}(\mathbf{x})$ is well-defined and unique.

By the optimality condition of $\widehat{\mathbf{x}}$, it is clear that $f(\widehat{\mathbf{x}}) + \frac{1}{2\gamma}\|\widehat{\mathbf{x}} - \mathbf{x}\|^2 \leq f(\mathbf{x})$, which proves the first inequality $f(\widehat{\mathbf{x}}) \leq f(\mathbf{x})$. Note that when $\mathbf{x} = \mathbf{x}_* \in \arg\min_{\mathbf{z}} f(\mathbf{z})$, we can prove that $\widehat{\mathbf{x}} = \mathbf{x} = \mathbf{x}_*$. This is because that by definition of $\widehat{\mathbf{x}}$, it is unique and satisfies $0 \in \partial f(\widehat{\mathbf{x}}) + \frac{1}{\gamma}(\widehat{\mathbf{x}} - \mathbf{x})$. When $\mathbf{x} = \mathbf{x}_*$, we have $0 \in \partial f(\mathbf{x})$ and $\mathbf{x}_*$ clearly satisfies the first-order condition $0 \in \partial f(\mathbf{x}_*) + \frac{1}{\gamma}(\mathbf{x}_* - \mathbf{x})$. By the uniqueness of $\widehat{\mathbf{x}}$, it follows that $\widehat{\mathbf{x}} = \mathbf{x}_*$.

The first-order condition $0 \in \partial f(\widehat{\mathbf{x}}) + \frac{1}{\gamma}(\widehat{\mathbf{x}} - \mathbf{x})$ also gives that $\operatorname{dist}(0, \partial f(\widehat{\mathbf{x}})) \leq \frac{1}{\gamma}\|\widehat{\mathbf{x}} - \mathbf{x}\|$. Also, we have $\nabla f_\gamma(\mathbf{x}) = \frac{1}{\gamma}(\widehat{\mathbf{x}} - \mathbf{x})$ (Rockafellar, 1970), which implies the second and the third inequality in (3).

