# OpenReview forum: "Universal  Stagewise Learning for Non-Convex Problems with  Convergence on  Averaged Solutions"
_ICLR.cc/2019/Conference_

### Official Review · AnonReviewer3 · 2018-10-18
**An interesting attempt trying to analyze the practical learning rate setting of SGD**

**Rating:** 6
**Confidence:** 4

**Review:**

In the paper, the authors try to analyze the convergence of stochastic gradient descent based method with stagewise learning rate and average solution in practice. The paper is very easy to follow, and the experimental results are clear. The following are my concerns:

1. In function (3), for any x in R^d, if \hat x  = prox_\gamma f (x), then f(\hat x ) <= f(x). This inequality looks not correct to me. If x = argmin_x f(x), the above inequality is obviously wrong.  It looks like that function (3) is a very important basis for the whole paper.

2. By using the weakly convex assumption and solving f_s, the authors transform a nonconvex nonsmooth problem to a convex problem. However, the paper didn't mention how to select \gamma in the algorithm. This parameter is nontrivial, if you set a small value, the problem is not convex and the analysis does not hold. In the experiment, the authors tune \gamma from 1 to 2000, which means that u < 1 or u < 1/2000.  Given neural network is a u-weakly convex problem or u-smooth problem, the theory does not match the experiment.

3. The authors propose a universal stagewise optimization framework and mention that the stagewise ADAGRAD obtains faster convergence than other analysis. My question is that, if it is a generic framework, how about the convergence rate for other methods? is there also acceleration for SGD or momentum SGD?

---

> ### Author Response · Authors · 2018-11-13
> **clarification of (3)**
>
> It is indeed correct.  Please note that when x=argmin_x f(x), we have \hat x= x and  f(\hat x) = f(x). Then the inequality f(\hat x)\leq f(x) is still correct. We have provided a proof of (3) in the Appendix of the revision (Appendix H on page 22), though it has been proved in earlier works. Please take a look.

---

> ### Author Response · Authors · 2018-11-13
> **regarding the generic framework**
>
> We did analyze different methods in our framework including SGD (section 4.1), stochastic heavy-ball method (also known as stochastic momentum method) (corresponding to \rho =0 in Algorithm 4, see Section 4.3), and stochastic Nesterov accelerated gradient method (corresponding to \rho=1 in Algorithm 4, see Section 4.3). Indeed, we provide a general convergence theory in Theorem 1 such that any suitable stochastic convex optimization algorithms can be analyzed. For example, for AMSGRAD (a variant of Adam with theoretical guarantee for convex optimization), we can derive a similar convergence result (i.e., 1/epsilon^4 iteration complexity) for our framework employing AMSGRAD as the subroutine. Other methods can be also analyzed in our framework (e.g., RMSProp (Mukkamala & Hei 2018)).  We are not clear what does the reviewer mean by acceleration of SGD or momentum SGD. Indeed, this is first work that establishes non-asymptotic convergence of stagewise momentum SGD (similar to algorithms used in practice for deep learning) for non-smooth non-convex problems.
>
> Mahesh Chandra Mukkamala, Matthias Hein. Variants of RMSProp and Adagrad with Logarithmic Regret Bounds. ICML 2018.

---

> ### Author Response · Authors · 2018-11-13
> **Thanks R3 for the review**
>
> Dear Reviewer 3,
>
> Thanks for reviewing our paper.
>
> We would like to request you to read our responses that clarify your concerns.  To summarize our responses: (i) the inequality (3) is indeed correct; (ii) we did analyze the algorithms employing SGD, and momentum-based SGD; (iii) the value  \gamma is selected based on the validation performance, which is a standard approach for setting parameters.
>
> Please take them into account when making the final recommendation. Great Thanks!
>
> Regards
> Authors

---

> > ### Comment · AnonReviewer3 · 2018-11-20
> > **Thanks for authors' clarification**
> >
> > Dear authors,
> >
> > I have read your rebuttals, and the following is my feedback:
> > 1) The inequality is correct.
> > 2) My concern is resolved.
> > 3) Although you tune the \gamma as per the validation error, I still don't think the algorithm and analysis in the paper match the experiment. In practice, people do not make deep learning problem convex at each stage.  In spite of this, it is still an interesting paper that tries to analyze the stage-wise learning rate scheme.
> >
> > Overall, I will upgrade my score to 6.

---

### Official Review · AnonReviewer2 · 2018-11-01
**Comparison over related work should be clarified. Measure of convergence rate should be justified**

**Rating:** 6
**Confidence:** 4

**Review:**

Non-convex optimization is a hot topic since many machine learning problems can be formulated as non-convex problems. In this paper, the authors propose a universal stage-wise algorithm for weakly convex optimization problems. The idea is to add a strongly convex regularizer centered at an iterate of previous stage to the objective function. This builds a convex function which can be optimized by any standard methods in the convex optimization setting. The authors developed convergence rates in expectation in terms of the gradient of envelope. Empirical results are also reported to show the effectiveness of the method.

Comments:

(1) The weakly-convex concept considered in this paper is very similar to the bounded non-convexity considered in the paper (Natasha: Faster Non-Convex Stochastic Optimization Via Strongly Non-Convex Parameter) (not cited). In particular, the Natasha paper also developed a multi-stage algorithm for bounded non-convexity optimization problems by adding strongly-convex regularizers centered at iterates of previous stages. The authors should discuss more extensively the related work to clarify their novelty.

(2) The convergence rate is measured by $\nabla\phi_\gamma(x_\tau)$. However, according to (3) , this only guarantees an upper bound on $\text{dist}(0,\partial\phi_\gamma(\text{prox}_{\gamma\phi_\gamma}(x_\tau)))$. The output of the algorithm is $x_\tau$ instead of $\text{prox}_{\gamma\phi_\gamma}(x_\tau)$. Is it possible to derive an upper bound on $\text{dist}(0,\partial\phi_\gamma(x_\tau))$?

---

> ### Author Response · Authors · 2018-11-13
> **Clarification of difference from Natasha and the convergence measure**
>
> Thanks for the valuable comments.
>
> Q1: Missing reference (Natasha).
> A: We have included the discussion about Natasha in the revision (the end of Related Work on page 3). We agree that both papers use the idea of adding a strongly convex regularizer to the objective function. However, this is a commonly used technique. It dates back to the proximal point method proposed in 1970s (e.g., Rockafellar (1970)). The recent works that use this idea for non-convex optimization include Carmon et al. (2016), Allen-Zhu (2017), Lan & Yang (2018) for smooth problems, and Davis & Grimmer (2017) for non-smooth problems. We have discussed the later work in the original submission. In the revision, we add the discussion about other works that add strongly convex regularizer to the objective. The key differences between our paper and the Natasha paper of Allen-Zhu is summarized below:
> a.	First, Allen-Zhu considers finite-sum problems, and assume the objective function has a smooth component. In contrast, we consider more general stochastic problems without assuming the function is smooth. Please check the Ex. 2 on page 5 for an example of non-smooth and non-convex functions, for which our algorithm is applicable but Natasha is not applicable.
> b.	Due to the strong condition (i.e., finite-sum structure and smoothness) made in the Natasha paper, they are able to get better complexity in terms of epsilon. However, in this paper we focus on how to explain the success of heuristic used in practice for solving deep learning problems, including stagewise step size, averaging and adaptive step size. Our theory covers most commonly used stochastic algorithms used in practice.
>
> Rockafellar, R. T. (1970). Convex analysis. Princeton: Princeton University Press.
> Yair Carmon, John C. Duchi, Oliver Hinder, and Aaron Sidford. Accelerated methods for non-convex optimization, arXiv, 2016.
> Guanghui Lan and Yu Yang. Accelerated stochastic algorithms for nonconvex finite-sum and multi-block optimization. CoRR, abs/1805.05411, 2018.
>
>
> Q2:   About the choice of the convergence measure.
> A. First,  please note that when the objective function is non-smooth (that is considered in this paper), it is challenging for an iterative algorithm to find a solution x_t such that dist(0, \partial f(x_t))\leq \epsilon. We have given one example in the paper (see the paragraph after eq. (3)). Consider min |x|, for an iterative algorithm that produces a non-optimal solution x_t (that is not zero), then  dist(0, \partial f(x_t)) is always 1. Indeed, this observation has been reported in several previous papers for non-smooth and non-convex optimization (Davis & Drusvyatskiy, 2018a; Drusvyatskiy & Paquette, 2018; Davis & Grimmer, 2017).  To address this issue, the convergence measure based on the Moreau envelope’s gradient is used following these papers, which ensures that the found solution x_t is very close to a solution that is epsilon stationary.
>
> Second, the good news. Actually, when the objective function is smooth, the upper bound of the the Moreau envelope’s gradient’s norm can be translated to an upper bound of the (projected) gradient’ norm that is commonly used as a convergence for smooth functions. Please see eqn. (4) and (5) and texts around them in the revision. It means that in the smooth case (which is a special case of weakly convex), the convergence of the |\nabla f_\gamma(x_\tau)| indeed transfers to a convergence of |\nabla f(x_\tau)|.

---

> > ### Comment · AnonReviewer2 · 2018-11-21
> > **Thanks for the detailed response**
> >
> > The authors' response clarify the difference between this work and the Natasha paper. My concern is addressed.

---

### Official Review · AnonReviewer1 · 2018-11-02
**Novel idea, Like the paper**

**Rating:** 8
**Confidence:** 4

**Review:**

Summary:
The paper presents an analysis and numerical evaluation of stagewise SGD, ADAGRAD and Stochastic momentum methods for solving stochastic non-smooth non-convex optimization problems.

Comments:
I find the ideas presented in this paper very interesting. The convergence analysis seems correct and the paper is reasonably well written, and tackles an important problem.

The analysis holds for μ-weekly convex functions. This assumption is really important for the development of the algorithm and the proposed analysis. I like the fact that the authors provide two examples showing that popular objective functions in machine learning satisfy this assumption.

The numerical evaluation is adequate showing the effectiveness  of the proposed stagewise algorithms.  However i have the follow suggestions/minor comments:

1) It will be nice to have also some plots showing the performance of the proposed method on the ImageNet dataset.
2) Another possible nice experiment will be a comparison of the four stagewise methods (SGD,ADAGRAD,SHB,SNAG) on the same dataset. Which one behaves better?

Minor Comments:
1) The captions of the figures can be more informative (mention also the division by column). First column is SGD, Second column Adagrad, etc.
2) Typos:
Section 1, last bullet point, second line: "stagwise"
Section 5, second paragraph , first line :"their their"
page 8, 3 line from the bottom:  "seems, indicate"

2) Missing reference.
In the area of stochastic gradient methods with momentum many papers have been proposed recently for the case of convex optimization that worth to be mentioned:
Gadat, Sébastien, Fabien Panloup, and Sofiane Saadane. "Stochastic heavy ball." Electronic Journal of Statistics 12.1 (2018): 461-529.
Loizou, Nicolas, and Peter Richtárik. "Momentum and stochastic momentum for stochastic gradient, Newton, proximal point and subspace descent methods." arXiv preprint arXiv:1712.09677 (2017).
Lan, Guanghui, and Yi Zhou. "An optimal randomized incremental gradient method." Mathematical programming (2017): 1-49.

Overall, I suggest to accept this paper.

---

> ### Author Response · Authors · 2018-11-13
> **Thanks for liking our paper**
>
> Thanks for your interest and the valuable comments on our work.
>
> Q1:  It will be nice to have more experiments on the ImageNet data set.
> A: We are running experiments on the ImageNet dataset and expect to include the results in the final version.
>
> Q2:  Another possible nice experiment will be a comparison of the four stagewise methods.
> A: Indeed, we list a result in Table 1 in the Appendix to compare all methods in terms of testing error.  There is no clear winner depending on datasets and on whether regularization is added. But they have comparable results. We plot the curves of the four stagewise methods for both training error and testing error in the updated version (see Figure 6).
>
> Q3:   Missing reference and typos
> A: Thanks for the suggestions. We have corrected the typos in the revision. We will look into the referred papers of stochastic momentum methods carefully and include them in appropriate places in the final version.

---

### Meta-Review · Area_Chair1 · 2018-12-12
**ICLR 2019 decision**

**Confidence:** 4
**Recommendation:** Accept (Poster)

**Metareview:**

This paper develops a stagewise optimization framework for solving non smooth and non convex problems. The idea is to use  standard convex solvers to iteratively optimize a regularized objective with penalty centered at previous iterates - which is standard in many proximal methods. The paper combines this with the analysis for non-smooth functions giving a more general convergence results. Reviewers agree on the usefulness and novelty of the contribution. Initially there were concerns about lack of comparison with current results, but updated version have addressed this issue.  The main weakness is that the results only holds for \mu weekly convex functions and the algorithm depends on the knowledge of \mu. Despite this limitations, reviewers believe that the paper has enough new material and I suggest for publication. I suggest authors to address these issues in the final version.